# Discovery of ultrafast spontaneous spin switching in an antiferromagnet by femtosecond noise correlation spectroscopy

M. A. Weiss [1], A. Herbst [1], J. Schlegel[1], T. Dannegger [1], M. Evers[1], A. Donges[1], M. Nakajima[2], A. Leitenstorfer [1], S. T. B. Goennenwein [1], U. Nowak [1] & T. Kurihara [1,3] ✉

Owing to their high magnon frequencies, antiferromagnets are key materials for future high-speed spintronics. Picosecond switching of antiferromagnetic spin systems has been viewed a milestone for decades and pursued only by using ultrafast external perturbations. Here, we show that picosecond spin switching occurs spontaneously due to thermal fluctuations in the antiferromagnetic orthoferrite $Sm_{0.7}Er_{0.3}FeO_3$. By analysing the correlation between the pulse-to-pulse polarisation fluctuations of two femtosecond optical probes, we extract the autocorrelation of incoherent magnon fluctuations. We observe a strong enhancement of the magnon fluctuation amplitude and the coherence time around the critical temperature of the spin reorientation transition. The spectrum shows two distinct features, one corresponding to the quasi-ferromagnetic mode and another one which has not been previously reported in pump-probe experiments. Comparison to a stochastic spin dynamics simulation reveals this new mode as smoking gun of ultrafast spontaneous spin switching within the double-well anisotropy potential.

Some of the most intriguing effects in physics rest on fluctuations. Incoherent thermal fluctuations of spins critically determine the magnetic properties of correlated systems. Thermally excited magnons are a driving force of a rich variety of fundamental physical phenomena such as phase transitions and spin caloritronic effects[1–3], whereas their non-deterministic properties promise unique applications such as probabilistic computing[4]. Incoherent spin fluctuations have traditionally been studied either indirectly through the temperature dependence of macroscopic properties such as heat capacity, conductivity, and magnetic susceptibility[5,6], or in the frequency domain through optical probes relying on, e.g., the Raman effect[7,8] or diffraction[9]. For measuring relatively slow spin fluctuation dynamics in paramagnets in the range of MHz to GHz frequencies, spin noise spectroscopy (SNS)[10–15] has been employed. In contrast to paramagnets, correlated spin systems exhibit collective excitations, i.e.,

magnons. Antiferromagnets (AFM) possess especially high-frequency magnons reaching into the THz range[16], and are thereby attracting enormous attention from the viewpoint of accelerating the conventional ferromagnet-based spintronics devices[17,18]. However, due to their ultrafast dynamics that go beyond state-of-the-art electronics, conventional SNS cannot detect antiferromagnetic spin fluctuations. The rich spin physics in AFMs arising from their complicated spin textures have only been resolved with ultrafast pump-probe spectroscopy, where time resolutions down to femto- or even attoseconds[19] are available. Still, such pump-probe measurements are of perturbative nature and therefore, they cannot detect the incoherent dynamics that spontaneously exist due to thermal or quantum mechanisms.

In this work, we experimentally demonstrate the spontaneous incoherent sub-THz magnon fluctuation dynamics in the AFM. This is achieved by a unique experimental principle, inspired by the emerging

[1]Department of Physics, University of Konstanz, D-78457 Konstanz, Germany. [2]Institute of Laser Engineering, Osaka University, 565-0871 Osaka, Japan. [3]The Institute for Solid State Physics, The University of Tokyo, 277-8581 Kashiwa, Japan. ✉e-mail: takayuki.kurihara@issp.u-tokyo.ac.jp

field of sub-cycle quantum optics[20–22]. Here, we analyse magnon fluctuation dynamics via their temporal autocorrelation function, by measuring the statistical correlations of polarisation noise imprinted on two subsequent femtosecond probe pulses (see Fig. 1a). The two linearly polarised, spectrally separate pulses with a variable time delay $\Delta t$ are focused on the sample. Upon transmission of the pulses at times $t$ and $t + \Delta t$, transient magnetisation fluctuations $\delta M_c(t)$ and $\delta M_c(t + \Delta t)$ parallel to the propagation direction introduce polarisation changes $\delta \alpha(t)$ and $\delta \alpha(t + \Delta t)$ to each probe, respectively, via the Faraday effect. The polarisation states of the transmitted probe pulses are individually analysed with independent polarimetric detectors. The pulse-to-pulse fluctuations of the detector output are extracted by sub-harmonic lock-in amplification[20], multiplied in real time and averaged over ~$10^8$ pulses at each delay position. By this method, the time correlation trace of the out-of-plane sub-THz magnetisation dynamics $\langle \delta M_c(t) \delta M_c(t + \Delta t) \rangle$ is precisely unravelled[22,23] (see "Methods" section for further details).

## Results

### Experimental sample

Our sample is a single crystal of the orthoferrite $Sm_{0.7}Er_{0.3}FeO_3$[24]. In this material, the electron spins of the $Fe^{3+}$ ions are antiferromagnetically coupled. A Dzyaloshinskii–Moriya interaction[25,26] (DMI) results in a slight spin canting and a weak net ferromagnetic moment (net magnetisation **M**). Orthoferrites have two exchange modes with resonance at multi-THz frequencies and two magnon modes in the sub-THz regime[27,28]. The latter include a quasi-ferromagnetic mode (qF mode) and a quasi-antiferromagnetic mode (qAF mode). The qF mode is characterised by a precession of the weak ferromagnetic moment around its equilibrium axis, whereas the qAF mode results in its longitudinal oscillation. $Sm_{0.7}Er_{0.3}FeO_3$ shows a temperature-induced second-order spin reorientation transition (SRT) close to room temperature[29], in which the magnetic anisotropy (but not the antiferromagnetic spin order) change. In the SRT region ($T_L < T < T_U$), **M** continuously rotates from along the crystallographic $a$-axis at the lower critical temperature $T_L$ to the $c$-axis at the upper critical temperature $T_U$. The critical temperatures have been previously measured to be $T_{L,ref} = 310$ K and $T_{U,ref} = 330$ K[24]. The SRT is expressed by a change in the free energy potential (see Fig. 1b). $T_L$ marks the temperature where the anisotropy difference between $a$- and $c$-axis disappears, causing an enhanced magnetic susceptibility in the $a$-$c$-plane and strong softening of the qF mode resonance frequency[24,30]. When such a system is thermally populated, one expects a strong enhancement of the angular distribution width at $T_L$.

### Volume scaling of the magnon noise

First, we investigate the magnon noise dynamics as a function of the time delay $\Delta t$ and its scaling with the probed volume $\Omega$ at a temperature of 294.15 K (Fig. 2a). The confocal position of the sample is identified by the maximum signal amplitude ($z = 0\ \mu m$, green graph). A systematic decrease of the amplitude is observed with extending $z$ up to $\pm 20\ \mu m$ (blue, yellow, magenta, and red graphs in Fig. 2a). The signals are symmetric around $\Delta t = 0$, consistent with the fact that we probe an autocorrelation of the temporal dynamics of the system. All waveforms exhibit a distinct peak at $\Delta t = 0$ followed by a gradual decrease and slow oscillations around the zero level that lasts for several tens of picoseconds. Figure 2b shows the noise amplitude at $\Delta t = 0$ as a function of the longitudinal sample position with respect to the optical focus (blue circles). As illustrated by the black graph, the correlated Faraday noise amplitude is fitted well by a function which is inversely proportional to the volume $\Omega(z)$ probed by the fundamental

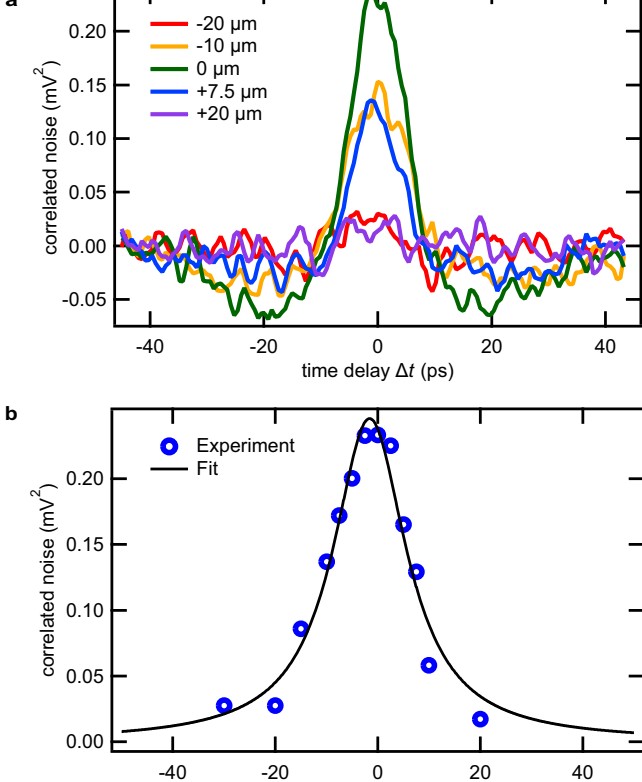

**Fig. 2 | Dependence of magnon noise dynamics on probing volume. a** Spin noise waveforms recorded at a constant temperature of 294.15 K for multiple longitudinal sample positions relative to the laser focus. The confocal position $z = 0$ was determined by maximising the amplitude at $\Delta t = 0$ (green graph). Amplitudes decrease monotonically with increasing distance from focus (blue, yellow, magenta and red graphs). **b** Correlated noise amplitude at $\Delta t = 0$ as a function of lateral sample position relative to the focus (blue open circles). The longitudinal position dependence was fitted with the function given in the Methods section.

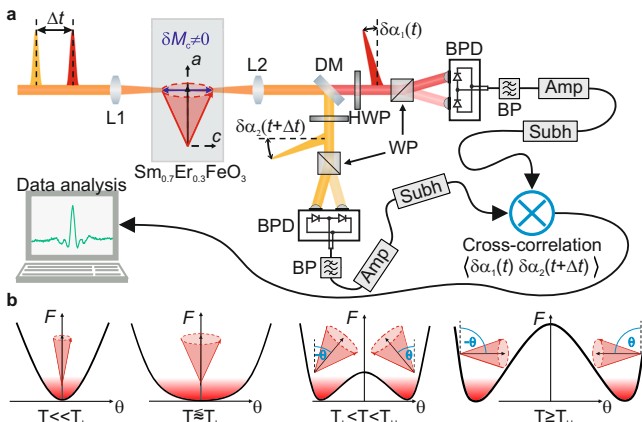

**Fig. 1 | Schematic illustration of the experimental setup and spin system. a** Due to the Faraday effect, two spectrally separated fs pulses (orange and red) of variable time delay $\Delta t$ experience a polarisation rotation proportional to out-of-plane spin fluctuations $\delta M_c$. Corresponding rotation angles $\delta \alpha_{1,2}$ are measured in separate elipsometers. After extraction of the pulse-to-pulse fluctuations from each branch, the cross-correlation function $\langle \delta \alpha_1 \delta \alpha_2 \rangle$ is calculated in real time as a function of the delay time $\Delta t$. L1,L2: lenses; DM: dichroic mirror; HWP: half-wave plate; WP: Wollaston prism; BPD: balanced photodetector; BP: electronic bandpass filter; Amp: transimpedance amplifier; Subh: sub-harmonic demodulation scheme. **b** Temperature evolution of the free energy $F$ and its influence on the spin noise dynamics (red cones) close to the spin reorientation in $Sm_{0.7}Er_{0.3}FeO_3$. The weak ferromagnetic moment **M** (black arrow) gradually rotates from $\theta = 0°$ at $T \leq T_L$ to $\theta = \pm 90°$ at $T \geq T_U$. Here, $\theta$ is the angle between **M** and the $a$-axis of the sample.

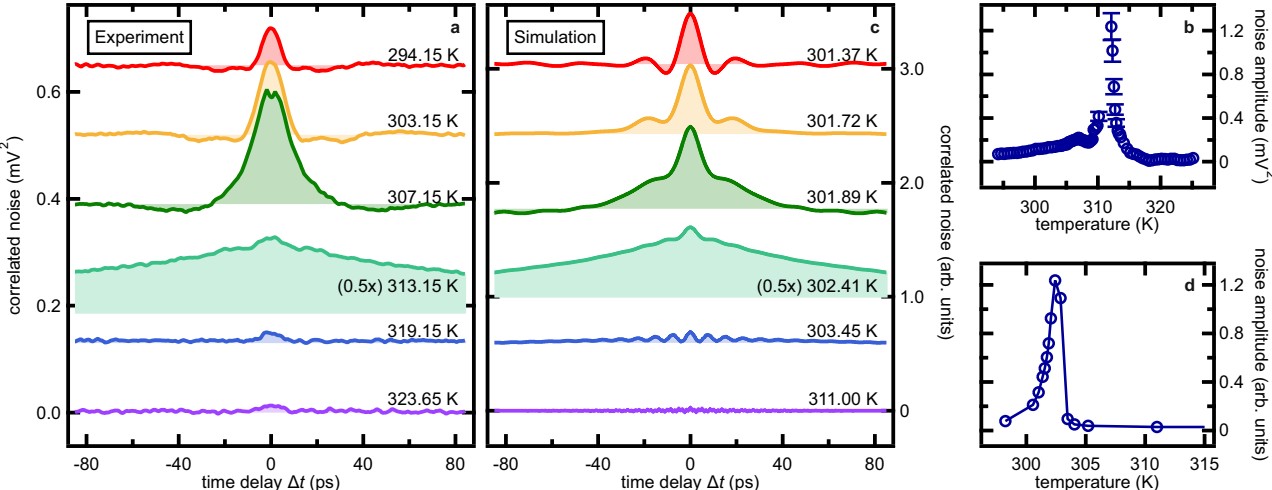

**Fig. 3 | Ultrafast magnon noise dynamics near spin reorientation in Sm₀.₇Er₀.₃FeO₃. a** Correlated noise as a function of time delay $\Delta t$ between probing pulses for multiple temperatures near the spin reorientation in $Sm_{0.7}Er_{0.3}FeO_3$. **b** Experimentally determined time-zero amplitude as a function of temperature. In the region of noise enhancement around 312 K, error bars are added considering the uncertainty associated with the background subtraction procedure (see "Methods" section). **c** Magnon noise simulation based on atomistic spin models and the stochastic Landau-Lifshitz-Gilbert equation. **d** Temperature evolution of the simulated time-zero amplitude.

Gaussian mode $\langle \delta\alpha(t)^2 \rangle \propto \frac{1}{\Omega(z)}$. Note that in paramagnets, the amplitude $\delta\alpha$ of the statistical fluctuations of $N$ independent spins within the probing volume $\Omega$ is known to follow the scaling law $\delta\alpha \propto \frac{1}{\sqrt{\Omega}} \propto \frac{1}{\sqrt{N}}$[11,15]. Here, we find the same volume scaling of Faraday noise as in conventional SNS. Note that this dependence is not trivially understood for correlated spin systems where individual spins are coupled to form collective magnons. Our observation indicates the existence of mutually incoherent oscillators whose spatial extent is smaller than the probe spot size. While the origin of such oscillators cannot be fully identified at this stage, we speculate that this scaling may be indicative of a multi-domain magnetic state that effectively limits the coherence length of magnons. Nevertheless, in the following, we fix our sample position to $z = 0\ \mu m$ to maximise the signal based on this feature.

**Temperature dependence of the waveforms around the SRT**

Figure 3a depicts the striking variation of spin noise autocorrelation waveforms found around the SRT. The amplitudes, oscillation periods and lifetimes depend strongly on temperature. Since in the SRT only the orientation of the spin system changes, this strong variation of the signal indicates that the observed autocorrelation signal truly comes from magnetisation noise dynamics. The finite width of the autocorrelation around the peak ($\Delta t = 0$) indicates that the system exhibits non-zero coherence on the ultrafast timescale. The oscillations with a period of tens of picoseconds, clearly visible in some of the autocorrelation waveforms (e.g., 294.15 K and 303.15 K) on both sides of the peak, reflects the spin precession mode. The temporal dynamics is discussed later in more detail.

Now, we first focus on the temperature evolution of the signal amplitude. The amplitude at $\Delta t = 0$ is depicted in Fig. 3b. A sharp amplitude increase is observed in the region around 312.15 K. This point is slightly higher but close to the estimated lower threshold temperature $T_L \sim 305\ K$ of the SRT in our sample, around which temperature the anisotropy softening results in an enhanced magnetic susceptibility[27]. Beyond this temperature, the noise amplitude decreases continuously, almost disappearing around the upper threshold $T_U = 320\ K$. The sharp decrease observed towards the higher temperature side is explained by the equilibrium rotation of the spin system within the SRT. In this temperature region, the net magnetisation **M** continuously rotates from **M** // $a$ (in-plane) to **M** // $c$ (out-of-plane) (see Supplementary Information SI2). Our Faraday probe is sensitive only to the $c$-axis magnetisation fluctuation $\delta M_c$ of the qF

mode, which is expected to reduce as $\delta M_c \propto \cos(\theta)$ towards higher temperature. Therefore, the noise amplitude is expected to decrease continuously. These temperature dependencies indicate that the amplitude of the observed magnon noise can be naively understood as the angle distribution of spins due to thermal occupation of the anisotropy potential well by magnons, consistent with the model described in Fig. 1b.

These findings are analysed exploiting simulations of the spin noise around the SRT in a generic orthoferrite with parameters fitting the equilibrium properties observed experimentally (see "Methods" section). The theory is based on an atomistic spin model and the stochastic Landau-Lifshitz-Gilbert equation[31,32]. Figure 3c depicts the simulated $c$-axis magnon noise autocorrelation trace in a 192 x 192 x 192 orthoferrite spin lattice. The simulation reproduces the temporal shape of the noise waveforms in Fig. 3a, including the symmetry around $\Delta t = 0$ and the temperature evolution of both the time-zero peak amplitude and the subsequent slow oscillation. The peak amplitude of the calculated waveforms is shown as a function of temperature in Fig. 3d. In the simulations, the SRT manifests as a strong noise enhancement around 302 K followed by a decrease at higher temperatures. The nearly quantitative agreement between the theoretical calculation and the experimental data allows us to investigate the stochastic nature of the spin noise dynamics from a microscopic viewpoint in the following discussions.

**Spectral analysis of magnetisation fluctuations**

We now analyse the dynamics in the frequency domain. The Fourier spectra of the detected noise autocorrelation waveforms are shown in Fig. 4a. Interestingly, two distinct spectral peaks (purple and green arrows) are observed for most temperatures. The frequencies and amplitudes of both peaks are strongly dependent on temperature. While the two peaks are clearly distinguishable for $T \leq 304\ K$, they exhibit similar frequencies when approaching the noise enhancement region around $T_L$ and eventually become indistinguishable due to the strong broadening. At temperatures $T \gg T_L$, the spectral amplitude is significantly reduced because of the SRT. Figure 4b shows the frequencies of each peak as a function of temperature. Both peaks experience a strong frequency reduction around $T_L \approx 305\ K$, which closely resembles the softening behaviour of the qF mode around the SRT. The temperature dependence of the high-frequency (HF) peak (green full circles in Fig. 4b) is shown to quantitatively match with

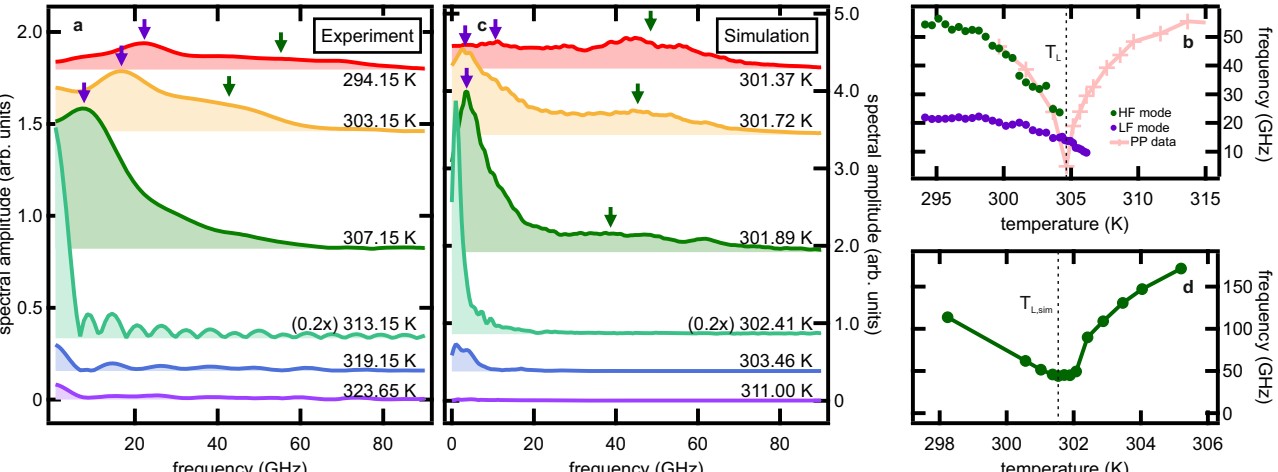

**Fig. 4 | Magnon noise spectra near SRT in Sm$_{0.7}$Er$_{0.3}$FeO$_3$. a** Fourier spectra of the measured magnon noise waveforms from Fig. 3a. Two maxima are resolved for $T < 307.15$ K (purple and green arrows), while only the low-frequency (purple arrows) peak prevails for higher temperatures. **b** Temperature evolution of the high-frequency peak (dark green full circles) and the low-frequency peak (purple full circles). The values were obtained either by fitting the spectra in **a** with a double Lorentzian function or evaluation of the 2nd derivative zero crossing points (see "Methods" section). For comparison, quasi-ferromagnetic magnon mode (qF mode) frequency data obtained by THz pump-near IR probe[24] are shown as pink crosses. The pump-probe data is shifted by −7.5 K to compensate for the different amount of stationary heating in our experiment. **c** Fourier spectra of the simulated waveforms (Fig. 3c). Green and purple arrows indicate the centre frequencies of the qF mode and the low-frequency feature, respectively. **d** Temperature evolution of the simulated c-axis projection of the qF mode frequency. The values were obtained by fitting the spectra in **c** with a triple Lorentzian function (see "Methods" section).

pump-probe data[24] (pink crosses), clearly identifying this HF peak as the qF mode in Sm$_{0.7}$Er$_{0.3}$FeO$_3$. On the other hand, the low-frequency peak (LF peak) has no correspondence in pump-probe measurements. It suggests that this LF peak is linked to the randomness of the spontaneous magnetic dynamics, which has now become accessible by the stochastic nature of our technique.

The Fourier spectra of the stochastic LLG simulations are depicted in Fig. 4c. The appearance of the two peaks and their softening around $T_{L,sim} \approx 301.5$ K (Fig. 4d) matches the experimental results in Fig. 4b. This agreement between the temperature dependence of the simulated qF mode fluctuation and the HF mode seen in the experiment further supports our assignment to the original qF mode. Conversely, the LF feature appears in the experimental data from the spectrum recorded at a temperature of $T = 294.15$ K to temperatures well beyond $T_L \approx 305$ K, whereas it is observed in a narrower temperature region $T \gtrsim T_{L,sim}$ in the simulation (Fig. 4c). This quantitative discrepancy stems from the practical limitations of our simulation in finding a set of parameters to fit the experimental results (see "Methods" section). Still, both the experimental and simulated temperature dependence of the LF spectral amplitude (Suppl. Figs. 1 and 4) follow a similar trend as the time-zero amplitude as a function of temperature shown in Fig. 3b. This finding suggests the underlying LF dynamics to contribute significantly to the total noise amplitude (Fig. 3b, d).

## Comparison with simulated temporal evolution of magnetisation

To gain insights into the physical origin of the LF feature, we now investigate the simulation data in more detail. Figure 5a shows results for the temporal evolution of the c-axis projection of the normalised magnetisation $m_c/m_S$ ($m_S$ is the magnetisation at saturation). For $T < T_{L,sim}$, the equilibrium axis of the normalised magnetisation is parallel to the a-axis. Consequently, fluctuations of $m_c/m_S$ centred around the origin are observed. When approaching $T_{L,sim}$, the fluctuations increase in amplitude and oscillation period in agreement with our previous discussion. For $T \gtrsim T_{L,sim}$, $m_c/m_S$ switching between two discrete states with similar amplitude but different sign (up- and down-state) become prominent, resembling random telegraph noise (RTN)[33–35] on a picosecond timescale (see Supplementary

Information SI5). With increasing temperature, the number of observed switches gradually decreases, while at the same time, the distance $\Delta m_c$ between the up-and down states increases. For sufficiently high temperatures $T \gg T_{L,sim}$, no more switching events are recorded. Here, $m_c/m_S$ always fluctuates around a preferred state. When comparing the temperatures at which the LF peak is observed in the simulated spectra (Fig. 4c) with the temperatures at which RTN is recorded in the magnetisation time traces (Fig. 5a), it becomes clear that the emergence of the LF feature is inherently linked to the RTN.

Note that the autocorrelation of a pure two-level random telegraph noise follows an exponential decay of the form $\left(\frac{\Delta m_c}{2}\right)^2 e^{-2\frac{|\Delta t|}{\tau}}$ for finite time $\Delta t$[36], with $\tau$ being the mean dwell time. Since both $\Delta m_c$ and $\Delta t$ increase with the progress of SRT, the RTN dynamics should contribute to an exponentially decaying component with an increasing amplitude and decay time. This behaviour is indeed observed in both the simulated and measured autocorrelation traces (Fig. 3a, c), which unambiguously proves that the measured noise signals reflect the RTN dynamics. In the frequency domain, it manifests itself as the LF peak. It should be mentioned that the Fourier transform of a purely exponentially decaying autocorrelation function results in a Lorentzian spectrum centred around zero[36,37]. In contrast, in our observations the LF feature exhibits a peak at finite frequencies. We attribute this difference to the limited time window over which our traces are analysed.

To investigate this RTN behaviour in further detail, we plot the trajectory of the LLG-simulated magnetisation for different temperatures around the SRT (Fig. 5b–f and Supplementary Video 9). At $T = 298.24$ K $< T_{L,sim}$ (Fig. 5b), the equilibrium magnetisation points along the a-axis ($\theta = 0$) and qF mode noise can be observed in the b- and c-projections (see Suppl. Fig. 5).

At $T = T_{L,sim} \approx 301.54$ K (Fig. 5c), an enhancement of the qF mode noise is observed. The spin system then rotates towards finite angle $\theta$ for $T > T_{L,sim}$. The $\pm\theta$ states are energetically degenerate and switching between them occurs for temperatures slightly above $T_{L,sim}$, i.e., for $T = 302.88$ K $\gtrsim T_{L,sim}$ (Fig. 5d). This results in an additional magnetisation noise contribution (see Suppl. Fig. 5c) and accounts for the strongly enhanced noise amplitude in the $T = 302.41$ K $\gtrsim T_{L,sim}$ waveform in Fig. 3c. The configuration of the sublattice magnetisation vectors and the net magnetisation for the

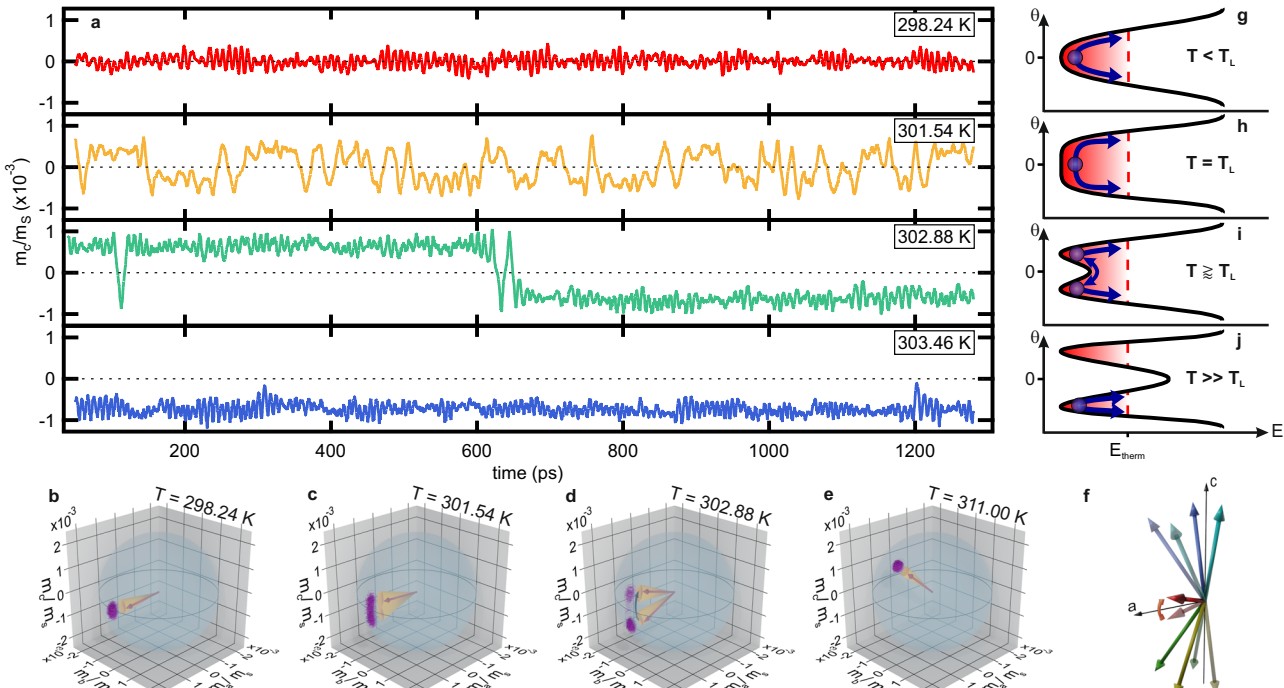

**Fig. 5 | Picosecond random switching in Sm$_{0.7}$Er$_{0.3}$FeO$_3$. a** Simulated time traces of the *c*-axis component $m_c$ of the magnetisation normalised to the saturation magnetisation $m_S$ for multiple temperatures near spin reorientation in Sm$_{0.7}$Er$_{0.3}$FeO$_3$. The first 50 ps are not shown, because here the system still equilibrates from the initial conditions. **b**–**e** Simulated trajectories (purple lines) of the normalised magnetisation for multiple temperatures near spin reorientation in Sm$_{0.7}$Er$_{0.3}$FeO$_3$. At temperature $T \gtrsim T_{L,sim}$ (**d**) switching events are recorded in the *c*-direction. The precession cones of the qF mode magnon are indicated in yellow. **f** Illustration of the sublattice magnetisation vectors in Sm$_{0.7}$Er$_{0.3}$FeO$_3$ magnetisation (blue, mint, green, yellow) and the net magnetisation (red) for the energetically degenerate $\pm\theta$ states. The canting angle of the sublattice magnetisation vectors and the thus resulting net magnetisation is highly exaggerated for visibility. **g**–**j** Orthoferrite potential landscape for different temperatures across the SRT. The red-dotted line indicates the thermal energy $E_{\text{therm}}$ of the system. At temperatures $T < T_L$ (**g**), the potential exhibits a parabolic shape and the fluctuations are restricted to a single minimum around $\theta = 0$. Slightly above $T_L$ (**h, i**), a double-well potential develops and the particle randomly switches between the minima located at $\pm\theta$. For $T_U > T \gg T_L$ (**j**), the energy barrier between the minima is larger than the thermal energy of the system and no more switching events occur.

switching $\pm\theta$ states is shown in Fig. 5f. As the temperature is elevated even further (Fig. 5e), the switching probability becomes lower until no more switching occurs. Here, the up-state is always preferred because of the initial conditions of the simulation. When approaching the upper threshold temperature $T_{U,sim}$, switching is observed in the *a*-projection due to anisotropy softening along this direction. Beyond $T_{U,sim}$, the equilibrium magnetisation becomes parallel to the *c*-axis (not shown) and the qF mode contribution to the noise vanishes (see Suppl. Figs. 2 and 4).

The physical picture of the RTN dynamics can be clearly understood by a model considering the stochastic switching between two energetically degenerate quasi-equilibrium states, which manifest as $\pm\theta$ rotation states of the equilibrium magnetisation due to the SRT (see Fig. 5g–j). In Sm$_{0.7}$Er$_{0.3}$FeO$_3$, the free energy density exhibits a parabolic shape for $T < T_L$, whereas for $T > T_L$ it evolves into a double-potential well with minima at $\pm\theta$[24]. The rotation angle $\theta$ and the height of the potential barrier separating the two minima gradually increase until $T = T_U$[38]. Random switching between $\pm\theta$ states occurs when the height of the potential barrier is low compared to the thermal energy of the system (Fig. 5i, $T \gtrsim T_L$). Further increase of the temperature in the order of 1 K significantly changes the barrier height, while thermal energy only changes marginally. As a result, the average lifetime $\tau$ on each quasi-equilibrium state increases, and the switching probability declines. Note that this model even reproduces our observation that the temperature at which the time-zero amplitude of the autocorrelation becomes maximal (Fig. 3b, d) is slightly higher than $T_L$ (Fig. 4b, d).

The evident connection between RTN in the simulated time traces and the LF peak in the spectra firmly establishes picosecond RTN as the physical origin of the experimental LF feature. It should be noted that stochastic physical systems exhibiting RTN have lately gathered prominence as a key enabler for probabilistic computing. For this purpose, systems showing high-frequency RTN are desired to implement faster computing times and higher precision[33–35]. While RTN on electronic timescales was intensively studied for decades in systems exhibiting charge carrier traps, e.g., in commercial semiconductor structures[39,40], the fastest RTN device reported so far remained in the nanosecond regime exploiting in-plane magnetic tunnel junctions[41,42]. To the best of our knowledge, the picosecond RTN reported here marks the record-switching speed to date. We attribute the high rate in Sm$_{0.7}$Er$_{0.3}$FeO$_3$ to the magnon frequency in the sub-THz region which results in shorter intervals between switching attempts as compared to conventional ferromagnets[42,43]. This result further highlights the capability of ultrafast SNS as a unique tool for probing the stochastic dynamics near a magnetic phase transition.

In summary, we demonstrate the time-domain observation of sub-THz magnon fluctuations in the antiferromagnetic orthoferrite Sm$_{0.7}$Er$_{0.3}$FeO$_3$ near the SRT. The drastic increase of amplitude and coherence time within the SRT together with the low-frequency peak observed in the spectrum is direct evidence of ultrafast random spin switching between two equilibrium states of the magnetic free energy. The random spin switching speed reported here is the fastest ever marked and may provide a key ingredient for ultrafast probabilistic computing operating at THz frequencies. Our experimental concept is not only limited to orthoferrites but also applicable to a wide range of correlated magnetic systems. Furthermore, our results shed new light on THz magnonics in AFMs, where the influence of incoherent spin dynamics has largely been dismissed. In future works, combining the

proposed concept with coherent excitations is expected to enable the seamless observation of spin dynamics from thermal equilibrium to nonequilibrium states.

## Methods
### Experiment and data post-processing
This study exploits a modelocked Er:fibre laser system emitting pulses of a width of 150 fs, a central wavelength of 1.55 μm and a total energy of 5 nJ at a repetition rate of 40 MHz. This output is frequency doubled in a periodically poled lithium niobate (PPLN) crystal, reaching a transparency window of the orthoferrites[44]. Subsequently, the frequency-doubled beam is spectrally split by a dichroic mirror, resulting in two linearly polarised, spectrally distinct femtosecond pulse trains. The central wavelengths of the two beams are 767 nm and 775 nm with 3 to 4 nm bandwidths, respectively. One of the probe pulses is time-delayed by $\Delta t$ with an optical delay stage. After spatial recombination by another dichroic mirror, the pulses are focused to a spot diameter below 2 μm on the orthoferrite sample with a transmissive objective lens of a numerical aperture of 0.4. The sample is a $d = 10$ μm thick, $c$-cut plate of single-crystal $Sm_{0.7}Er_{0.3}FeO_3$. The two probe pulses are linearly polarised along the $a$-axis. Upon transmission of the pulses through the sample of thickness $d$ at times $t$ and $t + \Delta t$, transient magnetisation fluctuations $\delta M_c(t)$ parallel to the propagation direction of the pulse trains introduce polarisation noise $\delta \alpha(t) \propto V(\lambda) \cdot d \cdot \mu_0 \cdot \delta M_c(t)$ via the magneto-optic Faraday effect, where $V(\lambda)$ is the wavelength dependent Verdet constant and $\mu_0$ is the vacuum permeability. After collimating them with an additional lens, the two probe beams are spatially separated with a dichroic mirror and sent into individual analysers to monitor their polarisation. Each detector consists of a half-wave plate (HWP), a Wollaston prism and a pair of balanced photo diodes. The angles of the HWPs are set to compensate for stationary components of the polarisation rotation induced by, e.g., biaxial birefringence of the sample[45]. The Wollaston prisms (WP) separate p- and s-polarised components of the probes. The intensity difference of the polarisation components is then detected in separate balanced photodetectors (BPD) and amplified with transimpedance amplifiers (Amp), respectively. Subsequently, the signals pass a 20 MHz bandpass filter (BP) and are demodulated at the first sub-harmonic of the laser repetition rate (20 MHz) with a radio-frequency lock-in amplifier (UHFLI, Zurich Instruments)[21]. The outputs of these demodulation channels now comprise the polarisation noise amplitudes $\delta \alpha(t)$ and $\delta \alpha(t + \Delta t)$, respectively, as well as uncorrelated components dominated by the shot noise of the photons in the probes. In a last step, both demodulation signals are multiplied in real-time inside the lock-in amplifier. This product is averaged over approximately $10^8$ pulse pairs per time delay $\Delta t$ to effectively yield the cross-correlation of Faraday noise $\langle \delta \alpha(t) \delta \alpha(t + \Delta t) \rangle$. In this way, we extract the tiny portion of correlated fluctuations originating from the sample response out of a much larger uncorrelated background. At the same time, the two-colour scheme avoids detrimental interference effects at short delay times, providing sensitive access to high frequencies. Furthermore, it enables the two probe beams to collinearly overlap before the objective lens, allowing for the beam spots on the sample surface to maximally mode matched. This is crucial to obtain magnon correlation signals with sufficiently strong amplitude at measurable levels.

Contrary to previous works[21], the two-colour scheme also allows us to sample the Faraday rotation with two separate BPDs. By this way, our time resolution is not limited by the deadtime of the photodetector, and we can exploit the full sub-picosecond resolution given by the duration of the probe pulses. This fact enables us to analyse the changes of spin noise over a magnetic phase transition in a correlated magnetic material with significantly shorter spin dephasing times.

The raw waveforms are post-processed with a third-order Savitzky-Golay filter for smoothing, as well as a linear baseline subtraction, where the average values of the first and last 5 ps serve as a reference. Close to the SRT, the combined effects of magnon softening and thermally induced random switching result in the magnetisation noise not fully decaying to zero within the observation window. In this case, the abovementioned linear baseline subtraction cannot be used without distorting the waveform. From 307.15 K upwards, we therefore use the average baseline of all waveforms recorded below 307.15 K as the reference for our linear baseline subtraction. Furthermore, in the SRT region above 307.15 K, a strong baseline increase is observed, the amplitude of which is strongly temperature dependent and most prominent at around 311 K. The baseline increase is asymmetric around $\Delta t = 0$ and therefore must result from external factors, such as a slight misalignment of the delay stage. In the temperature region between 310.35 K and 311.95 K, no signature of correlated noise was observed because of the large asymmetric background. Hence no meaningful evaluation could be carried out and we consequently neglect the data in this region in our discussion. In all other reliable datasets recorded for temperatures larger than 307.15 K where a slight asymmetric background is superimposed with the correlated spin noise, a slope correction is employed to remove the asymmetry. To account for potential artefacts in the correlated spin noise introduced by this procedure, we assign a relative uncertainty of 10% of the amplitude determined at each time delay.

### Atomistic spin model simulations
$Sm_{0.7}Er_{0.3}FeO_3$ is modelled as a generic orthoferrite with magnetic moments on the Fe sites only. These are treated as classical vectors $\mathbf{S}_i$ on a simple cubic lattice with four sublattices. The rare earth moments order only at very low temperatures of typically $T < 6$ K and are, hence, neglected. In orthoferrites, the nearest neighbour exchange constant $J_1$ is of the order of $-20$ meV leading to an antiferromagnetic order with a Néel temperature in the range of 630 K, whereas the next nearest neighbour exchange constant $J_2$ is much smaller and of the order of $-1$ meV[46–48]. A reorientation transition can be modelled by different thermally stable anisotropies, as it is done in ref. 49. Here the reorientation transition is due to a competition of second-order on-site anisotropy, favouring the [001] direction, and a second-order two-site anisotropy, favouring the [001] plane. The low-temperature state is dominated by the on-site anisotropy whereas the high-temperature state is dominated by the more thermally stable two-site anisotropy. With these two anisotropies, one would obtain a first-order reorientation transition. By adding a small cubic anisotropy preferring the [111] direction, one obtains a second-order reorientation transition in agreement with experiments. The strength of the cubic anisotropy determines the width of the reorientation transition. The weak ferromagnetism caused by the canting of the antiparallelly aligned sublattice magnetisations originates from the oxygen-mediated Dzyaloshinskii–Moriya interaction (DMI).

Consequently, the Hamiltonian of our model reads

$$H\{\mathbf{S}_i\} = -\sum_{\langle i,j \rangle} J_1 S_i^\upsilon S_{j\upsilon} - \sum_{\langle i,j \rangle} J_2 S_i^\upsilon S_{j\upsilon} - \sum_{\langle i,j \rangle} \varepsilon_{\upsilon\eta\lambda} D_{ij}^\upsilon S_i^\eta S_j^\lambda$$
$$- \sum_{\langle i,j \rangle} \kappa^{\upsilon\eta} S_{i\upsilon} S_{j\eta} - \sum_i \kappa_2^{\upsilon\eta} S_{i\upsilon} S_{i\eta} - \sum_i \kappa_4^{\upsilon\upsilon\eta\eta} S_{i\upsilon} S_{i\upsilon} S_{i\eta} S_{i\eta} \tag{1}$$

using the Einstein notation where $i$ and $j$ denote the site indices and $\upsilon, \eta$ and $\lambda$ the Cartesian directions. The first two double sums correspond to the nearest neighbour interaction with $J_1 = -22.32$ meV and the next nearest neighbour interaction with $J_2 = -1.4$ meV. The DMI is included in the nearest neighbour shell with DMI vectors having a length of 0.18 meV for in-plane interactions and 0.25 meV for out-of-plane interactions. The directions of the DMI vectors can be obtained using the symmetry rules of ref. 26. In the Hamiltonian $D_{ij}^\upsilon$ corresponds to the $\upsilon$-component of the DMI vector describing the interaction between the spins on lattice site $i$ and $j$. The two-site anisotropy is also included in the first shell with $\kappa^{zz} = -0.1255$ meV, leading to an easy

$xy$-plane. The second-order on-site anisotropy is $\kappa_2^{zz} = 0.905$ meV, which results in an easy $z$-axis. There is also a small in-plane contribution with $\kappa_2^{xx} = \kappa_2^{yy} = 0.015$ meV and $\kappa_2^{xy} = \kappa_2^{yx} = -0.015$ meV. The latter contribution is not necessary for the reorientation transition but lifts the degeneracy of the spins in the $xy$-plane. The fourth-order on-site anisotropy is $\kappa_4^{xxyy} = \kappa_4^{xxzz} = \kappa_4^{yyzz} = 0.036$ meV. With these parameters, the model undergoes a reorientation transition between 302 K and 322 K, where the Néel vector rotates continuously from the $z$-direction to the $x\bar{y}$-direction and the magnetisation from the $x\bar{y}$-direction to the $z$-direction (see Supplementary Information SI2). Note that the $x, y$ and $z$ coordinate axes of the Hamiltonian are parallel or antiparallel to the connection lines of the iron atoms, but not parallel to the crystallographic axes of an orthoferrite $a,b,c$. The crystallographic unit vectors are given by:

$$\mathbf{e}_a = \frac{1}{\sqrt{2}}(\mathbf{e}_x - \mathbf{e}_y), \mathbf{e}_b = \frac{1}{\sqrt{2}}(\mathbf{e}_x + \mathbf{e}_y), \mathbf{e}_c = \mathbf{e}_z, \qquad (2)$$

so that the $a$-direction corresponds to the $x\bar{y}$-direction, the $b$-direction to the $xy$-direction and the $c$-direction equals the $z$-direction.

To simulate the dynamics of the magnetic moments, the stochastic Landau-Lifshitz-Gilbert equation[31,32] is used which reads:

$$\frac{d}{dt}\mathbf{S}_i = -\frac{\gamma}{\mu_S(1+\alpha_d^2)}\mathbf{S}_i \times (\mathbf{H}_i + \alpha_d\mathbf{S}_i \times \mathbf{H}_i) \qquad (3)$$

with

$$\mathbf{H}_i = -\frac{\partial H}{\partial \mathbf{S}_i} + \boldsymbol{\xi}_i \qquad (4)$$

and a thermal Gaussian white noise term with

$$\langle \boldsymbol{\xi}_i(t) \rangle = 0, \langle \xi_{iv}(t)\xi_{j\eta}(t') \rangle = \frac{2\mu_S\alpha_d k_B T}{\gamma_i}\delta_{ij}\delta_{v\eta}\delta(t-t') \qquad (5)$$

with the dimensionless damping parameter $\alpha_d = 0.0002$. The gyromagnetic ratio is set to the value of a free electron $\gamma_i = 1.76086 \times 10^{11} \frac{Hz}{T}$, the magnitude of the magnetic moments is $\mu_S = 3.66\mu_B$. On this basis, the time evolution of a system of $192^3$ spins is numerically calculated via the stochastic Heun method. Our main output is the magnetisation $\mathbf{m}(t)$, which reads:

$$\mathbf{m}(t) = \frac{1}{N}\sum_i \mathbf{S}_i. \qquad (6)$$

$N$ is the number of spins and $\mathbf{S}_i$ denotes the spin at lattice site $i$. Thus, $\mathbf{m}(t)$ is dimensionless and normalised to unity in the following. The spectral noise amplitude $P_\eta$ (spectral power density) is calculated via:

$$P_\eta(f) = \left\langle \frac{\left|m_\eta^T(f)\right|^2}{T} \right\rangle \qquad (7)$$

with the Fourier transform:

$$m_\eta^T(f) = \int_0^T m_\eta(t)e^{-i2\pi ft}dt. \qquad (8)$$

Furthermore, the spectral amplitude is averaged over 20 to 30 simulation runs. The time-dependent correlated noise amplitude (autocovariance) is determined by the inverse Fourier transform of the spectral noise amplitude, taking advantage of the Wiener-Khinchin theorem.

It should be noted that because of the complexity of our simulation, it is practically challenging to find a set of spin model parameters that precisely fit all experimental measurements simultaneously, like the Néel temperature, reorientation temperatures and eigenfrequencies. Furthermore, additional effects that should be present in the experiment, such as the influence of domain states and possible crystal defects, are not covered by our simulations. The quantitative discrepancy found in the simulation results from the experimental values, such as the SRT temperature region and the qF mode frequency, stems from the above technical limitation.

### Z-scan
When a sample of thickness $d$ is placed into the focus of a Gaussian laser beam, the illuminated volume forms a position-dependent hyperboloid:

$$\Omega(z) = \pi w_0^2 \left[d + \frac{d^3/12 + dz^2}{z_R^2}\right], \qquad (9)$$

where $z$ is the position of the sample along the axis of light propagation relative to the laser focus, $w_0$ is the beam radius at its narrowest point and $z_R = \frac{\pi w_0^2}{\lambda}$ is the wavelength $\lambda$ dependent Rayleigh length. We estimate $z_R$ to be 7 μm. In conventional SNS, the statistical fluctuation of $N$ spins in the probing volume $\Omega$ results in Faraday noise of the order $\delta\alpha \propto \frac{1}{\sqrt{\Omega}}$. For a correlated noise amplitude, we hence expect $\langle\delta\alpha^2\rangle \propto \frac{1}{\Omega}$ at $\Delta t = 0$.

Inserting the formula for $\Omega(z)$ then yields the following fitting function for the correlated noise amplitude at $\Delta t = 0$ as a function of lateral sample position (Fig. 2b):

$$\frac{A}{\pi w_0^2 \left[d + \frac{d^3/12 + dz^2}{z_R^2}\right]} \qquad (10)$$

where $A$ is a proportionality constant.

### Spectra
The experimental noise waveforms are interpolated, zero-padded and smoothed. Subsequently, a FFT algorithm is harnessed to obtain high-resolution spectra.

The spectral features are analysed using a double Lorentzian peak fit. In spectra where the qF mode and the LF feature (see Fig. 4b) overlap, the centre frequencies are obtained by estimating the zero-crossings of the second derivative.

The simulated spectra are interpolated and smoothed, as well. The spectral features are then analysed using a triple Lorentzian peak fit where the peaks correspond to the LF feature, the qF mode and the qAF mode, respectively (see Suppl. Fig. 4a–c). We note that the experimental and simulated spectra are acquired through different analytical procedures. In the experiment, we directly measure the time-domain autocorrelation waveform and obtain the spectra via Fourier transforming after filtering and background subtraction as described above. On the other hand, in the simulation, we first calculate the magnetisation dynamics and Fourier transform them into the spectral domain. The spectra shown in Fig. 4c are the averages of 20 spectra simulated in such a way.

## Data availability
The datasets generated in the present study are available from the corresponding author upon reasonable request.

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

## Acknowledgements

This research was supported by the Overseas Research Fellowship of the Japan Society for the Promotion of Science (JSPS), Zukunftskolleg Fellowship from the University of Konstanz, JSPS KAKENHI (JP21K14550, JP20K22478, JP20H02206) and by the Deutsche Forschungsgemeinschaft (DFG, German Research Foundation)—Project-ID 425217212-SFB 1432. T.K. acknowledges Tohru Suemoto for fruitful discussion.

## Author contributions

T.K. and A.L. conceived the experiment. M.A.W., A.H. and T.K. developed the experimental system, performed the measurements, and analysed the data. J.S., T.D., M.E. and A.D. performed the numerical simulations under the supervision of U.N. M.N. produced the specimen. T.K., A.L. and S.T.B.G. co-supervised the project. M.A.W. and T.K. wrote the manuscript with the help of all co-authors.

## Funding

## Competing interests

The authors declare no competing interests.
