## [Peer Review File · Nature Communications]

Reviewers' Comments:

Reviewer #1:

Remarks to the Author:

The paper of M.A. Weiss and co-authors investigates the fluctuations of magnetization of the antiferromagnetic orthoferrite $\text{Sm}_{0.7}\text{Er}_{0.3}\text{FeO}_3$ by means of correlation measurements of sub-cycle probes that effectively perform a magneto-optic sampling.

I have carefully read the manuscript and studied the supplementary video and must say that it's been a long while since I hadn't read such an exciting, original and thorough paper. The paper brings novelties on many different levels and deserves being published in Nature Communications:

- It introduces, for the first time, magnetization noise correlation measurements by using sub-cycle probes that measure the magnetization along the optical axis via the magneto-optic effect. So far, only electro-optic correlation measurements have been reported, which were limited to electric field correlation. This entirely novel implementation of sub-cycle correlations opens an entire realm of studies that concentrate on spins and magnetization and is in my eyes a major advance. The use of tightly focused probes enables an entirely new way to characterize the spin fluctuations with extreme temporal and spatial resolution.
- The correlation data is convincing, with good signal-to-noise ratio, which allows the authors to study the spin fluctuations not only in time domain but also in frequency domain. They are able to do this on sub-picosecond and picosecond timescales which is a clear advantage over other noise measurements which are typically limited to microseconds or at most nanoseconds.
- It investigates the spin fluctuation dynamics of SmErFeO_3 around the phase transition where the energy landscape goes from a single well to a double well with different stiffnesses. The authors provide very compelling numerical calculations that match the experimental observations
- The authors provide numerically simulated evolution of the spin dynamics below, around and above the temperature-induced phase transition.

Given my assessment above, I have mainly a few minor comments and many questions – especially because it seems to me that the technique they propose can be very powerful to extract various properties of magnetic samples. I'd appreciate if the authors could elaborate briefly.

- It appears to me that the experimental and numerical data in Fig. 4a and c are labeled wrongly (experiment should be simulation and vice-versa).
- It would be great if the authors could elaborate a bit more on their assumption that inside their probed volume all spins are oscillating incoherently, and thereby their total amplitude scales like \sqrt{N} . It is not very intuitive how this compares to typical correlation lengths of magnon modes.
- Related to the question above, could one fancy an experiment where all spins are locked to each other e.g. by coherent driving or injection locking? What impact would that have on the measured correlation traces/spectra?
- The authors show convincing data how the peak of the correlated noise decreases when increasing the probed volume in Fig. 2a, but I don't fully grasp how the signal would change if instead one would make the probed volume smaller and smaller? Would a tip-based technique be – in principle – compatible with their measurements?
- Related to my question above, would it be possible to use correlation measurements of this type to characterize the size of domain walls?
- Did the authors also look at the direct measurement of the magnetization $\langle m(x) \rangle$ in addition to their correlation measurements? It appears to me that above the phase transition they should see a signature of the magnetization rotation. Is this correct?
- Did the authors consider measuring the correlations also at different positions in-plane i.e. $\langle \delta m(x) \delta m(x+dx) \rangle$? Could such measurement be useful to retrieve the scattering length of magnon modes in which the phases of spins are correlated? What would such measurement yield if the two probes were in two different domains?
- Related to dynamics of phase transition: When passing the phase transition back and forth, does one retrieve the same correlation curve below the transition temperature?
- In my view, somehow the title of the paper does not grasp fully the capabilities of the measurement technique compared to other techniques, but of course it is up to the authors to decide.

Reviewer #2:

Remarks to the Author:

Review of Manuscript „Ultrafast spontaneous spin switching in an antiferromagnet“ by M. A. Weiss et al.

In this paper a novel scheme to measure the autocorrelation of the incoherent magnon fluctuations in a canted antiferromagnet $\text{Sm}_{0.7}\text{Er}_{0.3}\text{FeO}$ was demonstrated (Fig. 1a). This scheme involves the measurement of the correlation of optical polarization fluctuations, spontaneously induced on the two time-delayed optical probe pulses by magnetic fluctuations via the Faraday effect in a weakly ferromagnetic $\text{Sm}_{0.7}\text{Er}_{0.3}\text{FeO}$ sample. For the purposes of their separate detection and analysis, these two optical probe pulses are spectrally separated.

It was suggested that as the material temperature approaches the spin reorientation transition (SRT), the magnetic fluctuations will increase in amplitude, and will also develop special temporal dynamics. This will affect the noise autocorrelation signal, whose dynamics and strength will thus be a fingerprint of the spontaneous SRT in the crystal (Fig. 1b).

A theory was developed based on the atomistic spin model and the LLG equation, yielding at the end the autocorrelation of the incoherent magnon fluctuation in a crystal. These theoretically computed autocorrelation signals were then compared with the experimentally measured ones, and the conclusion was reached that the model accurately describes the microscopic physics of the SRT. The theory, at this point assumed to be a correct one, was then further analyzed, and the theoretically calculated temperature-dependent magnetization dynamics was presented, indeed showing a spontaneous magnetization switch around a critical temperature (Fig. 5 in the manuscript). This is the main result of this work.

This work concerns a modern and very interesting topic of ultrafast magnetization dynamics, which is of both fundamental and theoretical importance. Spontaneous, incoherent effects such as fluctuations-driven phase transitions are indeed very difficult to observe directly in an ultrafast experiment performed at a certain repetition rate (not in a single-shot regime). In this respect, the idea behind this experiment is indeed an ingenious one: measurement of the noise autocorrelation function, which will contain the information about the incoherent fluctuations themselves. I find the experimental results definitely convincing. I particularly appreciate the transparency of the authors regarding the treatment of the measured autocorrelation traces such as presented in Fig.2, in particular regarding the subtraction of dispersive background and asymmetry removal in some of the measured signals.

Since the main result of this paper rests on the direct comparison of experimentally measured and theoretically computed magnetic noise autocorrelation functions, and on the further analysis of the theoretical results, the accuracy of the theory becomes important. In my opinion, the authors have presented sufficient indication that their model is qualitatively correct.

On the other hand, I would like to point out a rather significant discrepancy between the experimental and theoretical temperature ranges, where the SRT transition is supposed to occur. Whereas the measured noise autocorrelation dynamics, covering all structural magnetic phases, spans the temperature interval of 294.15 – 323.65 K, i.e. the range of 29.5 K, the theoretically calculated similar-looking dynamics spans the significantly, 3 times – narrower, temperature interval of 301.37 – 311.00 K, i.e. the range of 9.63 K (see Figs. 3a,c). The temperature positions of measured and calculated maximum fluctuations are also shifted with respect to one another (see Figs. 3b,d). Further, in the computed magnetization dynamics shown in Fig. 5, which is the main quantitative result of this paper, the calculated magnetic switching transition, covering all magnetic phases, occurs on an even narrower interval of only ca. 5 K, whereas in the experiment this seems to happen over the interval of ca. 30 K. The discrepancy between the data and the theory is mentioned by the authors rather in passing, but its effect on the accuracy of the main conclusion of this work is not sufficiently discussed. Indeed, would the computed switching dynamics in Fig. 5 still look the same or similar, if the whole transition would span a ca. 6-times larger temperature range, like it was observed in the experiment? This is the only point in this work, which I believe needs significant further clarification. Which parameters in the model would

need to be altered in order to achieve a better matching between the model and the data? Are the theoretical assumptions at hand sufficient to achieve better quantitative agreement, or additional physical processes need to be included? All this, in my opinion, deserves substantial and transparent discussion.

Additionally I would like to recommend the authors to reformulate the section in Methods, describing the production of the two time- and spectrally separated probe pulses at 770 nm and 780 nm. Reading this paragraph for the first time led me to think in the beginning that it was 1.55 μm and 775 nm pulses, that were further used as the probes.

Generally, I find this work interesting indeed, and the presented physical picture of the fluctuation-driven spin reorientation dynamics generally convincing. I believe that this work can be published in Nature Communications once the points regarding the quantitative discrepancy between the data and the model are properly addressed.

Reviewer #3:

Remarks to the Author:

Weiss et al. studied spin noise spectra (SNS) in antiferromagnetic orthoferrite and observed SNS enhancement at around the spin reorientation transition (SRT). This was studied by measuring cross correlation of Faraday rotation using two laser pulses with changing delay time. I have following questions and comments about the measurement results of SNS and the ultrafast spin switching in antiferromagnet. Overall, I feel that discussions of measurement/theoretical results are poor to conclude the main story of this manuscript.

1. As we can see in Fig. 3 and Fig. 4, the SNS at SRT becomes broad (sharp) in time-domain (frequency-domain). This means that correlation time becomes long at around SRT, which is opposite with the title of this manuscript. So, my question is how to evaluate spin switching time of double-well potential in the measurement shown in Fig. 3 and Fig. 4? Authors mentioned picosecond stochastic switching, however, there is no description about how to evaluate this switching time. Is stochastic switching really within picosecond timescale?

2. I would like authors to mention how to understand the spectral shape in Fig. 3a for example in F-mode, i.e., temperature is below SRT. What determines the width of the signal? If spin noise does not have a color, i.e. white noise, then signal becomes delta-function like. I could not understand the origin of shape from the text.

3. When we see Fig. 3a below SRT, i.e. 294.15 K, there is a large peak at zero delay time. In addition to this main peak, there is a small oscillating signal. But when we see in Fig. 4a below SRT, there is a main peak at around 20 GHz, which may be corresponding to a small oscillating signal in Fig. 3a. So, authors seem to perform some data analysis to obtain Fig. 4a from Fig. 3a to remove main peak in Fig. 3a, but this is not clearly mentioned in the text.

4. In the case of stochastic switching of ferromagnet, the decrease of double-well potential leads to increase of switching speed as the Neel-Arrhenius law. I feel that the results presented in this manuscript may also be understood by the Neel-Arrhenius law. For the present manuscript, I cannot understand whether the stochastic switching reported in this study stems from peculiar spin configuration in antiferromagnet or not. This is related to the novelty of the work. This must be discussed in the text.

In addition to above comments, I have following technical questions,

1. In SNS experiment, two color laser pulses are used and they have different wavelength may be 770 nm and 780 nm as written in the Methods. Can authors comment on why they use so close wavelength? Since femtosecond pulse laser with pulse width ~ 150 fs has bandwidth ~ 10 nm, I think that two laser pulses with close wavelength is difficult to be separated completely by the dichroic mirror.

2. In the calculation of resonance frequency is much higher than that obtained in the experiment [Fig. 4b and Fig. 4c]. What does this difference come from?

Reviewer #1 (Remarks to the Author):

Summary:

The paper of M.A. Weiss and co-authors investigates the fluctuations of magnetization of the antiferromagnetic orthoferrite $\text{Sm}_{0.7}\text{Er}_{0.3}\text{FeO}_3$ by means of correlation measurements of sub-cycle probes that effectively perform a magneto-optic sampling.

I have carefully read the manuscript and studied the supplementary video and must say that it's been a long while since I hadn't read such an exciting, original and thorough paper. The paper brings novelties on many different levels and deserves being published in Nature Communications:

- It introduces, for the first time, magnetization noise correlation measurements by using sub-cycle probes that measure the magnetization along the optical axis via the magneto-optic effect. So far, only electro-optic correlation measurements have been reported, which were limited to electric field correlation. This entirely novel implementation of sub-cycle correlations opens an entire realm of studies that concentrate on spins and magnetization and is in my eyes a major advance. The use of tightly focused probes enables an entirely new way to characterize the spin fluctuations with extreme temporal and spatial resolution.

- The correlation data is convincing, with good signal-to-noise ratio, which allows the authors to study the spin fluctuations not only in time domain but also in frequency domain. They are able to do this on sub-picosecond and picosecond timescales which is a clear advantage over other noise measurements which are typically limited to microseconds or at most nanoseconds.

- It investigates the spin fluctuation dynamics of SmErFeO_3 around the phase transition where the energy landscape goes from a single well to a double well with different stiffnesses. The authors provide very compelling numerical calculations that match the experimental observations

- The authors provide numerically simulated evolution of the spin dynamics below, around and above the temperature-induced phase transition.

Given my assessment above, I have mainly a few minor comments and many questions – especially because it seems to me that the technique they propose can be very powerful to extract various properties of magnetic samples. I'd appreciate if the authors could elaborate briefly.

- We greatly appreciate the reviewer's positive assessment and constructive suggestions. We believe that our approach will be highly attractive in many fields of study, including THz science, ultrafast magnetism, and statistical physics. In view of the reviewer's comments, we have added revisions to our manuscript, as shown in the following point-by-point replies to each comment.

Comments:

Comment 1-1

It appears to me that the experimental and numerical data in Fig. 4a and c are labeled wrongly (experiment should be simulation and vice-versa).

- The labels are correct in their original appearance (Fig. 4 a is experiment and c is theory). The spectra of the simulation look relatively noisy because of the different analytical procedures used for acquiring the spectra. In the experiment, we obtain the autocorrelation traces directly and Fourier transform them to acquire the spectra. The noise which is embedded in the time domain is mostly at relatively high frequencies (hundreds of gigahertz), therefore the background noise contributions to the investigated spectral region (<100 GHz) are small. Consequently, the displayed spectra are smooth in this frequency region. On the other hand, in the simulation, we first calculate the magnetization dynamics in the time-domain (such as e.g., Fig. 5) and Fourier transform them into the spectral domain. This is repeated for 20-30 simulation runs to smoothen the data and obtain the mean spectra (Fig. 4c). Finally, we inverse-Fourier transform these averaged spectra into the time domain to obtain the autocorrelation function. This has turned out to be the most efficient way for calculating autocorrelation functions from simulations. It should be noted that even in the averaged spectra some uncertainty remains (Fig. 4c). However, transformed back into the time domain (Fig. 3 c), such spectral modulation mostly leads to the emergence of temporal “pedestal” pulses at later times, and therefore does not have a significant influence within the time window plotted here. This is why the autocorrelation function calculated in the simulation look smooth. To conclude, in the experimental data the waveforms in the time domain are noisier than the spectra, whereas in the simulation the opposite trend is observed, as pointed out by the reviewer. This difference is due to different analytical protocols used in our study, as described in the Methods section and Supplementary Materials. It should be also noted that the number of statistics is different between the experiment and the simulation. In the experimental traces, we average over hundreds of millions of repetitions for one data point in the autocorrelation trace, whereas in the simulation, the waveform calculation is repeated by 20-30 times at maximum due to the calculation cost. This also leads to the quantitative difference in the signal to noise level between the experiment and the simulation. In response to this comment, we have added the following description in the manuscript:

End of Methods section:

- Before:
“The simulated spectra are interpolated and smoothed, as well. The spectral features are then analysed using a triple Lorentzian peak fit where the peaks correspond to the LF feature, the F mode and the AF mode, respectively (see Suppl. Fig 4a-c).”
- After:
“The simulated spectra are interpolated and smoothed, as well. The spectral features are then analysed using a triple Lorentzian peak fit where the peaks

correspond to the LF feature, the F mode and the AF mode, respectively (see Suppl. Fig 4a-c). We note that the experimental and simulated spectra are acquired through different analytical procedures. In the experiment, we directly measure the time-domain autocorrelation waveform and obtain the spectra via Fourier transforming after filtering and background subtraction as described above. On the other hand, in the simulation, we first calculate the magnetisation dynamics and Fourier transform them into the spectral domain. The spectra shown in Fig. 4c are the averages of 20 spectra simulated in such a way.”

Comment 1-2

It would be great if the authors could elaborate a bit more on their assumption that inside their probed volume all spins are oscillating incoherently, and thereby their total amplitude scales like \sqrt{N} . It is not very intuitive how this compares to typical correlation lengths of magnon modes.

- We would like to note that while the Z-scan result follows a clear $1/\sqrt{V}$ scaling, it does not directly mean that literally all the Fe^{3+} sublattice spins that constitute antiferromagnetism fluctuate randomly, as would be the case for paramagnets. Since the magnon mode is clearly observed, spatial coherence between the individual spins should exist in the more microscopic spatial dimension compared to our probe spot size. Strictly said, the $1/\sqrt{V}$ law only indicates the existence of some kind of *oscillators* that are smaller than the spot size and randomly vibrating without the mutual coherence.
- From the experimental dataset obtained so far, it is not possible to pin down the origin of such (mesoscopic) oscillators. Therefore, we would like to abstain from providing too much speculation in the main manuscript for the moment. However, one possible hypothesis may be the influence of the magnetic domain structure. Around the SRT, the magnetic domain size becomes small due to the reduced magnetic anisotropy, presumably much smaller than our laser spot size. Such domain structures should influence the coherence length of magnon waves. Under the existence of the domain walls, propagating magnons are expected to experience scattering, which effectively limits the maximum spatial coherence length within each small domain. Therefore, the stationary magnetic domains make the magnetization to fluctuate uniformly within each of the small domains but fluctuate incoherently relative to the distanced domains, leading to the observed $1/\sqrt{V}$ scaling law.
- In response to this comment, we have added the following explanation in the main text:

Main text, last of the paragraph describing Fig.2:

- Before:
“Note that this dependence is not trivially understood for correlated spin systems where individual spins are coupled to form collective magnons. Nevertheless, in the following we fix our sample position to $z = 0 \mu\text{m}$ to maximise the signal based on this feature.”
- After:
“Note that this dependence is not trivially understood for correlated spin systems where individual spins are coupled to form collective magnons. **Our observation**

indicates the existence of the mutually incoherent oscillators whose spatial extent is smaller than the probe spot size. While the origin of such oscillators cannot be fully identified at this stage, we speculate that this scaling may be indicative of a multi-domain magnetic state that effectively limits the coherence length of magnons.

Nevertheless, in the following we fix our sample position to $z = 0 \mu\text{m}$ to maximise the signal based on this feature.”

Comment 1-3

Related to the question above, could one fancy an experiment where all spins are locked to each other e.g. by coherent driving or injection locking? What impact would that have on the measured correlation traces/spectra?

- Thank you again for another interesting question. It can be imagined that the situation depends on the amplitude of the coherent driving. Let us consider the case wherein the magnons are driven prior to the probe pulses by, e.g., THz magnetic field pumping or optomagnetic excitation. If the amplitude of the induced spin precession is sufficiently small, the coherent and incoherent dynamics can be linearly superposed, causing only moderate or little change in the noise dynamics. However, when the excitation amplitude becomes large enough and the spin systems behave nonlinearly, the coherent and incoherent components can potentially mix with each other. In such a regime, there might be a chance that there appears, as the referee suggests, some fancy and interesting dynamics wherein the noises are synchronously modulated by the magnon oscillations (i.e., squeezing and antisqueezing, etc.). Since in many cases the nonlinear driving of magnon systems require the relatively intense laser pulses working with kHz repetition rates, it is a rather challenging task to directly combine it with our current scheme which works in the 10 MHz repetition ranges.
-
- In response to this comment, we have added the following remark at the end of conclusion in the main manuscript:

Main text, end of conclusion:

- Before:
“Our experimental concept is not only limited to orthoferrites but also applicable to a wide range of correlated magnetic systems. Furthermore, our results shed new light on THz magnonics in AFMs, where the influence of incoherent spin dynamics has largely been dismissed.”
- After:
“Our experimental concept is not only limited to orthoferrites but also applicable to a wide range of correlated magnetic systems. Furthermore, our results shed new light on THz magnonics in AFMs, where the influence of incoherent spin dynamics has largely been dismissed. **In future works, combining the proposed concept with coherent excitations is expected to enable the seamless observation of spin dynamics from thermal equilibrium to nonequilibrium states.**”

Comment 1-4

The authors show convincing data how the peak of the correlated noise decreases when increasing the probed volume in Fig. 2a, but I don't fully grasp how the signal would change if instead one would make the probed volume smaller and smaller? Would a tip-based technique be – in principle – compatible with their measurements?

- In relation to our reply to the first comment, the $1/\sqrt{V}$ scaling law is expected to hold and the signal increases until the probe spot size becomes comparable to the average size of the incoherent mesoscopic oscillator (probably domains and/or the magnon coherence length). In that sense, we agree that some kind of near-field techniques (plasmon structure) may be helpful in the future to decrease the probe volume and enhance the signal strength even further.

Comment 1-5

Related to my question above, would it be possible to use correlation measurements of this type to characterize the size of domain walls?

- In relation to the previous and the first comment, we envision that the domain structure may have some influence on the detected signals. At the same time, the domain *walls* are usually much smaller than the domain size by orders of magnitude (generally tens of nm to sub-micrometer). Therefore, we are not sure if it has made any strong contribution in the signals currently observed. However, there is a possibility that such information can be obtained by using a different type of probes that can detect much smaller volume of the sample.

Comment 1-6

Did the authors also look at the direct measurement of the magnetization $\langle m(x) \rangle$ in addition to their correlation measurements? It appears to me that above the phase transition they should see a signature of the magnetization rotation. Is this correct?

- The mean magnetization m_z along the *c*-axis direction is routinely monitored during the alignment procedures. It is detected in the RF lock-in amplifier by demodulating the photodiode signals with the original repetition rate of our laser system at 40 MHz, instead of the sub-harmonics 20 MHz which is used to measure the fluctuations. We use it as an indicator of the SRT, since in the SRT temperature region, the 40 MHz component tends to exhibit finite values because of the stationary magnetization m_z which remains due to the imperfect cancellation of the *+c*- and *-c*-tilted domains within the probed volume. This signal increases towards the higher temperature side because of the stationary rotation of magnetization towards *c* axis within the SRT.

Comment 1-7

Did the authors consider measuring the correlations also at different positions in-plane i.e. ? Could such measurement be useful to retrieve the scattering length of magnon modes in which the phases of spins are correlated? What would such measurement yield if the two probes were in two different domains?

- We appreciate this fascinating suggestion of the referee. In principle, measuring the two different probe positions should provide information related to the magnon coherence length. Unfortunately, our setup is currently not constructed in a way that the separation distance between the two probes can be continuously tuned. When the probe spot on the sample moves, it also changes the focus position of the outgoing probe pulses on the balanced photodiodes and thus necessitates the entire alignment of a balanced detector arm. However, this is simply a technical challenge and we believe that it can be resolved in the future by a descent mechanics. Nevertheless, we completely agree with the reviewer that the spatial correlation of the magnons in the spontaneous regime is one of the most interesting properties that is expected to be clearly distinct from the paramagnetic systems wherein no spatial correlation exists. We would love to tackle this point in future works.

Comment 1-8

Related to dynamics of phase transition: When passing the phase transition back and forth, does one retrieve the same correlation curve below the transition temperature?

- The key qualitative features presented in the manuscript, namely the enhancement of the amplitude and coherence time of the autocorrelation waveform, as well as the appearance of the two spectral components in their Fourier transform, always reproduce. However, the detailed features of the waveform vary slightly each time the thermal cycle is repeated. We ascribe this feature to the intrinsic variation of the magnetic environment in the probe spot due to static magnetic domains. Since the domain structure around the spot volume is expected to be randomly formed, it can influence the magnon dynamics in a random manner.

Comment 1-9

In my view, somehow the title of the paper does not grasp fully the capabilities of the measurement technique compared to other techniques, but of course it is up to the authors to decide.

- We appreciate the suggestion from the referee very much. In the original title we focused solely on the physics that has been discovered. However, since our work also deals intensively with the development of the new and broadly applicable measurement technology, we were intrigued to include some sense of technical achievement in the title. Therefore, we have modified the title of our manuscript in the following manner:

- Before:
“Ultrafast spontaneous spin switching in an antiferromagnet”
- After:
“Discovery of ultrafast spontaneous spin switching in an antiferromagnet by femtosecond noise correlation spectroscopy”

Reviewer #2 (Remarks to the Author):

Summary:

Review of Manuscript „Ultrafast spontaneous spin switching in an antiferromagnet“ by M. A. Weiss et al.

In this paper a novel scheme to measure the autocorrelation of the incoherent magnon fluctuations in a canted antiferromagnet $\text{Sm}_{0.7}\text{Er}_{0.3}\text{FeO}$ was demonstrated (Fig. 1a). This scheme involves the measurement of the correlation of optical polarization fluctuations, spontaneously induced on the two time-delayed optical probe pulses by magnetic fluctuations via the Faraday effect in a weakly ferromagnetic $\text{Sm}_{0.7}\text{Er}_{0.3}\text{FeO}$ sample. For the purposes of their separate detection and analysis, these two optical probe pulses are spectrally separated.

It was suggested that as the material temperature approaches the spin reorientation transition (SRT), the magnetic fluctuations will increase in amplitude, and will also develop special temporal dynamics. This will affect the noise autocorrelation signal, whose dynamics and strength will thus be a fingerprint of the spontaneous SRT in the crystal (Fig. 1b).

A theory was developed based on the atomistic spin model and the LLG equation, yielding at the end the autocorrelation of the incoherent magnon fluctuation in a crystal. These theoretically computed autocorrelation signals were then compared with the experimentally measured ones, and the conclusion was reached that the model accurately describes the microscopic physics of the SRT. The theory, at this point assumed to be a correct one, was then further analyzed, and the theoretically calculated temperature-dependent magnetization dynamics was presented, indeed showing a spontaneous magnetization switch around a critical temperature (Fig. 5 in the manuscript). This is the main result of this work.

This work concerns a modern and very interesting topic of ultrafast magnetization dynamics, which is of both fundamental and theoretical importance. Spontaneous, incoherent effects such as fluctuations-driven phase transitions are indeed very difficult to observe directly in an ultrafast experiment performed at a certain repetition rate (not in a single-shot regime). In this respect, the idea behind this experiment is indeed an ingenious one: measurement of the noise autocorrelation function, which will contain the information about the incoherent fluctuations themselves. I find the experimental results definitely convincing. I particularly appreciate the transparency of the authors regarding the treatment of the measured autocorrelation traces such as presented in Fig.2, in particular regarding the subtraction of dispersive background and asymmetry removal in some of the measured signals.

Since the main result of this paper rests on the direct comparison of experimentally measured and theoretically computed magnetic noise autocorrelation functions, and on the further analysis of the theoretical results, the accuracy of the theory becomes important. In my opinion, the authors have presented sufficient indication that their model is qualitatively correct.

- We thank Reviewer 2 very much for the positive comments and many valuable suggestions. In reply to the comments, we have made revisions on the manuscript as described in detail in the following.

Comments:

Comment 2-1

On the other hand, I would like to point out a rather significant discrepancy between the experimental and theoretical temperature ranges, where the SRT transition is supposed to occur. Whereas the measured noise autocorrelation dynamics, covering all structural magnetic phases, spans the temperature interval of 294.15 – 323.65 K, i.e. the range of 29.5 K, the theoretically calculated similar-looking dynamics spans the significantly, 3 times – narrower, temperature interval of 301.37 – 311.00 K, i.e. the range of 9.63 K (see Figs. 3a,c). The temperature positions of measured and calculated maximum fluctuations are also shifted with respect to one another (see Figs. 3b,d). Further, in the computed magnetization dynamics shown in Fig. 5, which is the main quantitative result of this paper, the calculated magnetic switching transition, covering all magnetic phases, occurs on an even narrower interval of only ca. 5 K, whereas in the experiment this seems to happen over the interval of ca. 30 K. The discrepancy between the data and the theory is mentioned by the authors rather in passing, but its effect on the accuracy of the main conclusion of this work is not sufficiently discussed. Indeed, would the computed switching dynamics in Fig. 5 still look the same or similar, if the whole transition would span a ca. 6-times larger temperature range, like it was observed in the experiment? This is the only point in this work, which I believe needs significant further clarification. Which parameters in the model would need to be altered in order to achieve a better matching between the model and the data? Are the theoretical assumptions at hand sufficient to achieve better quantitative agreement, or additional physical processes need to be included? All this, in my opinion, deserves substantial and transparent discussion.

- We agree that the simulations are indeed important for the understanding of our measurements. Let us start with a general comment: the numerical simulations of a reorientation transition based on an atomistic spin model is a very difficult task, with not too many successful examples published in the literature (one of which being our Ref.49, another has just been published, PRB 107, 184426 (2023)). The complications are manifold. Leading energies in the simulated Hamiltonian are exchange interactions. They determine the Neel temperature, while the DMI is responsible for the spin canting. The different anisotropy contributions determine the ground state spin orientation, the reorientation transition temperatures, the order of the phase transition, and its temperature range. But these anisotropy energies are two to three orders of magnitude smaller than the exchange energies and the reorientation transition is driven by a very delicate balance of the three different anisotropy energies, which are considered in our model. At the same time, all these parameters determine the resonance frequencies. We did our best to find a set of parameters that reproduces our experimental findings as close as possible. We simulate already an enormous system size of 192^3 spins. To obtain a single time-dependent magnetization curve as shown in Figure 5a, a simulation run takes about 72 hours with a highly optimized code running on a graphics card. Slight modification of some of the model parameters might indeed lead to a better quantitative agreement with some of the

experiments - but at the cost of others. It is practically impossible, to find a set of only three anisotropy parameters, that fits simultaneously all experimental measurements. And even a "perfect" set of model parameters could not do that, since some deviations have to be expected, because we cannot model all crystallographic details of our sample.

Nevertheless, our model fits the reorientation transition quite well. In fact, there are two critical points, the lower one where the rotation of the order parameter starts (about 302K), - and that is the one we investigate - and the upper one where the rotation ends (322K). In the experiment, the transition is between 310 and 330 K. So, the width of the transition region is identical and in absolute numbers the simulations are only about 3% off.

However, the referee is right that the range of temperatures we show in our Figs. 3 and 5 is much narrower in simulations than in the experiment. This is also rooted in the discrepancy in the anisotropy parameters and the resulting qFM frequency in our simulation. As obvious from the fact that the qFM frequency is somewhat higher in the calculation than in the experiment, the anisotropy potential is steeper in the simulation in most part of the calculated temperature region. This means that the wide angular distribution of spins by the thermal occupation of the potential well can occur only in temperature regions close to T_L , where the potential softens the most. At the same time, the other complexity which should be present in the experiment that potentially leads to the broadening of the temperature dependence, such as mixed static domain states within the probed spot and the crystal imperfections, are not considered in the simulation.

- The referee is right that these discrepancies were only mentioned in passing and we added the following sentence (highlighted) to the main text and in Methods section:

Main text, paragraph comparing the simulated spectrum with the experiment (Fig. 4)

- Before:
“The Fourier spectra of the stochastic LLG simulations are depicted in Fig. 4c. The appearance of the two peaks and their softening around $T_{L, \text{sim}} \approx 301.5$ K (Fig. 4d) matches the experimental results in Fig. 4b. This agreement between the temperature dependence of the simulated F mode fluctuation and the HF mode seen in the experiment further supports our assignment to the original F mode. Conversely, the LF feature appears in the experimental data from the spectrum recorded at a temperature of $T = 294.15$ K to temperatures well beyond $T_L \approx 305$ K, whereas it is observed in a narrower temperature region $T \gtrsim T_{L, \text{sim}}$ in the simulation (Fig. 4c). Both the experimental and simulated temperature dependence of the LF spectral amplitude (Suppl. Figs. 1,4) follow a similar trend as the time-zero amplitude as a function of temperature shown in Fig. 3b. This finding suggests the underlying LF dynamics to contribute significantly to the total noise amplitude (Figs. 3b,d).”
- After:
“The Fourier spectra of the stochastic LLG simulations are depicted in Fig. 4c. The appearance of the two peaks and their softening around $T_{L, \text{sim}} \approx 301.5$ K (Fig. 4d) matches the experimental results in Fig. 4b. This agreement between the

temperature dependence of the simulated F mode fluctuation and the HF mode seen in the experiment further supports our assignment to the original F mode. Conversely, the LF feature appears in the experimental data from the spectrum recorded at a temperature of $T = 294.15$ K to temperatures well beyond $T_L \approx 305$ K, whereas it is observed in a narrower temperature region $T \approx T_{L,\text{sim}}$ in the simulation (Fig. 4c). **This quantitative discrepancy stems from the practical limitations of our simulation in finding a set of parameters to fit the experimental results (see **Methods** section). Still, both the experimental and simulated temperature dependence of the LF spectral amplitude (Suppl. Figs. 1,4) follow a similar trend as the time-zero amplitude as a function of temperature shown in Fig. 3b. This finding suggests the underlying LF dynamics to contribute significantly to the total noise amplitude (Figs. 3b,d)."**

Methods section, end of subsection "Atomistic spin model simulations"

- Before:
"Furthermore, the spectral amplitude is averaged over 20 to 30 simulation runs. The time dependent correlated noise amplitude (autocovariance) is determined by inverse Fourier transform of the spectral noise amplitude, taking advantage of the Wiener-Khinchin theorem."
- After:
"Furthermore, the spectral amplitude is averaged over 20 to 30 simulation runs. The time dependent correlated noise amplitude (autocovariance) is determined by inverse Fourier transform of the spectral noise amplitude, taking advantage of the Wiener-Khinchin theorem.
It should be noted that because of the complexity of our simulation, it is practically challenging to find a set of spin model parameters that precisely fit all experimental measurements simultaneously, like the Neel temperature, reorientation temperatures and eigen frequencies. Furthermore, additional effects which should be present in the experiment, such as the influence of domain states and possible crystal defects, are not covered by our simulations. The quantitative discrepancy found in the simulation results from the experimental values, such as the SRT temperature region and the qFM frequency, stems from the above technical limitation."

Comment 2-2

Additionally I would like to recommend the authors to reformulate the section in Methods, describing the production of the two time- and spectrally separated probe pulses at 770 nm and 780 nm. Reading this paragraph for the first time led me to think in the beginning that it was 1.55 μm and 775 nm pulses, that were further used as the probes.

- Thank you for the suggestion. Indeed, this comment led us to the realization that the original description was confusing. We have now revised the following sentences to clarify our experimental setting:

Beginning of Methods section:

- Before

“This study exploits a modelocked Er:fibre laser system emitting pulses of a width of 150 fs at a central wavelength of 1.55 μm , repetition rate of 40 MHz and with total energy of 5 nJ. This output is frequency doubled in a periodically-poled lithium niobate (PPLN) device, reaching a transparency window of the orthoferrites⁴³. Subsequently, the pulses are spectrally split by a dichroic mirror, resulting in two linearly polarised, spectrally distinct femtosecond pulse trains with a time delay Δt provided by an optical delay stage in one branch. After spatial recombination by another dichroic mirror, the pulses are focused to a spot diameter of $<2 \mu\text{m}$ on the orthoferrite sample with a transmissive objective lens with a numerical aperture of 0.4. Central wavelengths are set to 770 nm and 780 nm, respectively, with polarisation along the a -axis of the sample. The sample is a $d = 10 \mu\text{m}$ thick, c -cut plate of single-crystal $\text{Sm}_{0.7}\text{Er}_{0.3}\text{FeO}_3$.”

- After

“This study exploits a modelocked Er:fibre laser system emitting pulses of a width of 150 fs, a central wavelength of 1.55 μm and a total energy of 5 nJ at a repetition rate of 40 MHz. This output is frequency doubled in a periodically-poled lithium niobate (PPLN) device, reaching a transparency window of the orthoferrites⁴⁴. Subsequently, the frequency-doubled beam is spectrally split by a dichroic mirror, resulting in two linearly polarised, spectrally distinct femtosecond pulse trains. The central wavelengths of the two beams are 767 nm and 775 nm with 3 to 4 nm bandwidths, respectively. One of the probe pulses is time delayed by Δt with an optical delay stage. After spatial recombination by another dichroic mirror, the pulses are focused to a spot diameter below 2 μm on the orthoferrite sample with a transmissive objective lens of a numerical aperture of 0.4. The sample is a $d = 10 \mu\text{m}$ thick, c -cut plate of single-crystal $\text{Sm}_{0.7}\text{Er}_{0.3}\text{FeO}_3$. The two probe pulses are linearly polarized along the a -axis.”

Comment 2-3

Generally, I find this work interesting indeed, and the presented physical picture of the fluctuation-driven spin reorientation dynamics generally convincing. I believe that this work can be published in Nature Communications once the points regarding the quantitative discrepancy between the data and the model are properly addressed.

- Thank you very much for the positive comment. We believe that the clarification on this point as presented in the above replies has further improved the quality of our manuscript.

Reviewer #3 (Remarks to the Author):

Summary:

Weiss et al. studied spin noise spectra (SNS) in antiferromagnetic orthoferrite and observed SNS enhancement at around the spin reorientation transition (SRT). This was studied by measuring cross correlation of Faraday rotation using two laser pulses with changing delay time. I have following questions and comments about the measurement results of SNS and the ultrafast spin switching in antiferromagnet. Overall, I feel that discussions of measurement/theoretical results are poor to conclude the main story of this manuscript.

- We thank Reviewer 3 for reading the manuscript and valuable comments. On the basis of the suggestions, we have made significant revisions on our main text. Especially, we have added a new section in the Supplementary material wherein the quantitative evaluation of the switching time is performed.

Comments:

Comment 3-1

As we can see in Fig. 3 and Fig. 4, the SNS at SRT becomes broad (sharp) in time-domain (frequency-domain). This means that correlation time becomes long at around SRT, which is opposite with the title of this manuscript. So, my question is how to evaluate spin switching time of double-well potential in the measurement shown in Fig. 3 and Fig. 4? Authors mentioned picosecond stochastic switching, however, there is no description about how to evaluate this switching time. Is stochastic switching really within picosecond timescale?

- Generally, the timescale of the stochastic switching (the mean dwelling time τ of the spin system in one of the potential minima) is directly reflected in the decay time of the autocorrelation peak around $\Delta t = 0$ [Balakrishnan, V. *Mathematical Physics: Applications and Problems* (SPRINGER NATURE, 2020)]. As the referee noted, this decay time significantly increases up to a couple of tens to hundred picoseconds within the SRT temperature range (see, e.g., experimental waveform at 313.15 K), due to the elevation of the barrier height separating the potential wells. It should be noted, however, that this component survives down to lower temperatures wherein the barrier height is much smaller (e.g., experimental waveform at 307.15 K). In such regions, the decay time is significantly shorter and is in the order 10-20 ps, which is completely consistent with our main statement that the spin switching is occurring in the picosecond time scales.

To quantitatively evaluate the dwelling time of the spins, we added a new chapter in the Supplementary Materials (“**Mean dwell time τ estimation of the random telegraph noise**”). Here, we compare the following two methodologies:

- (1) The number of switching events at each temperature is extracted from the simulated magnetization time traces showing RTN in the time domain on the basis of an established algorithm [Yuzhelevski, Y. Yuzhelevski, M. & Jung, G. Random telegraph noise analysis in time domain. *The Review of scientific instruments* 71, 1681-1688; 10.1063/1.1150519 (2000)], which is conventionally utilized for the time-domain investigation of the RTN dynamics. Using the number of switching events within the simulated time traces, the mean dwell time τ_{sim} is calculated.
- (2) The simulated autocorrelation traces were fitted by an exponential function $\left(\frac{\Delta m_c}{2}\right)^2 e^{-2\frac{\Delta t}{\tau}}$ to directly obtain τ as a fitting parameter.

As a result, both approaches gave very similar values, confirming the feasibility of the second approach (see Supplementary Materials). The excellent quantitative agreement between the above two analysis methods indicates that the dwelling time of the spin system can be indeed extracted by fitting the decay time of the $\Delta t = 0$ peak by an exponential function. On this basis, we then employed the exponential fitting to the experimentally detected autocorrelation waveforms. The result shows that the dwelling time at relatively low temperature, e.g., 307.15 K is $\tau_{\text{exp}} = 21.8$ ps. At the lowest evaluated temperature of 304.35 K, this value further reduced to $\tau_{\text{exp}} = 11.21$ ps.

This analysis quantitatively proves that the random switching of the $\text{Sm}_{0.7}\text{Er}_{0.3}\text{FeO}_3$ spin system we have investigated undoubtedly occurs in the picosecond timescale.

Furthermore, we have revised the main text in the following manner to explain the abovementioned physical picture in detail.

Main text, after Figure 4:

- Before:

“To gain insights into the physical origin of the LF feature, we now investigate the simulation data in more detail. Figure 5a shows results for the temporal evolution of the c -axis projection of the normalised magnetisation m_c/m_S (m_S is the magnetisation at saturation). For $T < T_{L,\text{sim}}$, the equilibrium axis of the normalised magnetisation is parallel to the a -axis. Consequently, fluctuations of m_c/m_S centred around the origin are observed. When approaching $T_{L,\text{sim}}$, the fluctuations increase in amplitude and oscillation period in agreement with our previous discussion. For $T \gtrsim T_{L,\text{sim}}$, m_c/m_S switching between two discrete states with similar amplitude but different sign (up- and down-state) become prominent, resembling random telegraph noise (RTN^{33–35}) on a picosecond time scale. With increasing temperature, the number of observed switches gradually decreases until no more switching events are recorded for $T \gg T_{L,\text{sim}}$. Here, m_c/m_S always fluctuates around a preferred state. When comparing the temperatures at which the LF peak is observed in the simulated spectra (Fig. 4c) with the temperatures at which RTN is recorded in the magnetisation time traces (Fig. 5a), it becomes clear that the emergence of the LF peak is inherently linked to the observation of picosecond RTN. It should be noted that Fourier transform of a RTN signal should result in an exponentially decaying autocorrelation function and thus in a Lorentzian spectrum centred around zero³⁶. In contrast, in our observations the LF mode exhibits a peak at finite frequencies. We attribute this finding to the limited time window over which our traces are analysed.”

- After:

“To gain insights into the physical origin of the LF feature, we now investigate the simulation data in more detail. Figure 5a shows results for the temporal evolution of the c -axis projection of the normalised magnetisation m_c/m_S (m_S is the magnetisation at saturation). For $T < T_{L, sim}$, the equilibrium axis of the normalised magnetisation is parallel to the a -axis. Consequently, fluctuations of m_c/m_S centred around the origin are observed. When approaching $T_{L, sim}$, the fluctuations increase in amplitude and oscillation period in agreement with our previous discussion. For $T \gtrsim T_{L, sim}$, m_c/m_S switching between two discrete states with similar amplitude but different sign (up- and down-state) become prominent, resembling random telegraph noise (RTN^{33–35}) on a picosecond time scale (Suppl. Fig 7). With increasing temperature, the number of observed switches gradually decreases, while at the same time, the distance Δm_c between the up-and down states increases. For sufficiently high temperatures $T \gg T_{L, sim}$, no more switching events are recorded. Here, m_c/m_S always fluctuates around a preferred state. When comparing the temperatures at which the LF peak is observed in the simulated spectra (Fig. 4c) with the temperatures at which RTN is recorded in the magnetisation time traces (Fig. 5a), it becomes clear that the emergence of the LF feature is inherently linked to the RTN.

Note that the autocorrelation of a pure two-level random telegraph noise follows an exponential decay of the form $\left(\frac{\Delta m_c}{2}\right)^2 e^{-2\frac{|\Delta t|}{\tau}}$ for finite time Δt ³⁶, with τ being the mean dwell time. Since both Δm_c and Δt increase with the progress of SRT, the RTN dynamics should contribute to an exponentially decaying component with an increasing amplitude and decay time. This behaviour is indeed observed in both the simulated and measured autocorrelation traces (Fig. 3a,c), which unambiguously proves that the measured noise signals reflect the RTN dynamics. In the frequency domain, it manifests itself as the LF peak. It should be mentioned that the Fourier transform of a purely exponentially decaying autocorrelation function results in a Lorentzian spectrum centred around zero^{36,37}. In contrast, in our observations the LF mode exhibits a peak at finite frequencies. We attribute this difference to the limited time window over which our traces are analysed. ”

Comment 3-2

2. I would like authors to mention how to understand the spectral shape in Fig. 3a for example in F-mode, i.e., temperature is below SRT. What determines the width of the signal? If spin noise does not have a color, i.e. white noise, then signal becomes delta-function like. I could not understand the origin of shape from the text.

- The peak width of an autocorrelation trace reflects the coherence time of the system. Naively said, it is a convoluted time of the magnon oscillation period and the mean dwell time associated with the switching event in our case. Specifically in the low-temperature phase (sufficiently below the SRT region, e.g., 294.15 K in the experiment or 301.37 K in the simulation) where the switching dynamics becomes less pronounced and the signal is dominated by the qFM magnons, the width of the $\Delta t = 0$ peak in the autocorrelation trace is

determined by the oscillation period and the damping timescale of the qFM magnon. Since the spin noise does have a well-defined qFM eigenmode, the autocorrelation trace exhibits the finite width near $\Delta t = 0$ as well as the coherent oscillation surviving within the damping time, as can be visible in both the simulated and experimentally obtained traces.

- To address this point more clearly, we added the following explanation in the beginning of the paragraph describing Fig. 3a:

Main text, description of Figure 3:

- Before:
“Figure 3a depicts the striking variation of spin noise autocorrelation waveforms found around the SRT. The amplitudes, oscillation periods and lifetimes depend strongly on temperature. To focus on the temperature evolution of the signal amplitude, the amplitude at $\Delta t = 0$ is depicted in Fig. 3b.”
- After:
“Figure 3a depicts the striking variation of spin noise autocorrelation waveforms found around the SRT. The amplitudes, oscillation periods and lifetimes depend strongly on temperature. **Since in the SRT only the spin system changes, this strong variation of the signal indicates that the observed autocorrelation signal truly comes from magnon noises. The width of the autocorrelation around the peak ($\Delta t = 0$) mainly reflects the coherence time of the magnon dynamics. The oscillations with period of tens of picosecond clearly visible in some of the autocorrelation waveforms (e.g., 294.15 K and 303.15 K) on both sides of the peak reflects the spin precession mode. The temporal dynamics is discussed later in more detail.**
Now, we first focus on the temperature evolution of the signal amplitude. The amplitude at $\Delta t = 0$ is depicted in Fig. 3b...”

Comment 3-3

3. When we see Fig. 3a below SRT, i.e. 294.15 K, there is a large peak at zero delay time. In addition to this main peak, there is a small oscillating signal. But when we see in Fig. 4a below SRT, there is a main peak at around 20 GHz, which may be corresponding to a small oscillating signal in Fig. 3a. So, authors seem to perform some data analysis to obtain Fig. 4a from Fig. 3a to remove main peak in Fig. 3a, but this is not clearly mentioned in the text.

- Apart from the data analysis procedures stated in the original manuscript and the Methods section (smoothing the autocorrelation waveforms by Savitzky-Golay filtering, zero padding, etc.), no further post processing is performed. The spectra in Figure 4 a are direct Fourier transform of the waveforms in Figure 3 a. Specifically in the data at 294.15 K mentioned by the Reviewer, the subsidiary oscillation on both sides of the $\Delta t = 0$ peak contains the negative values and it compensates for the main peak at $\Delta t = 0$, leaving little spectral weight at the zero frequency. The fast oscillation most clearly visible from ~ 10 ps until around 40 ps

corresponds to the 53 GHz peak (qFM). The 20 GHz component reflects the $\Delta t = 0$ main peak combined with the subsidiary negative parts on both sides lasting until $\sim \pm 30$ ps.

Comment 3-4

4. In the case of stochastic switching of ferromagnet, the decrease of double-well potential leads to increase of switching speed as the Neel-Arrhenius law. I feel that the results presented in this manuscript may also be understood by the Neel-Arrhenius law. For the present manuscript, I cannot understand whether the stochastic switching reported in this study stems from peculiar spin configuration in antiferromagnet or not. This is related to the novelty of the work. This must be discussed in the text.

The Neel-Arrhenius law suggested by the Reviewer is a relation that describes a thermally activated switching process in a bistable system where two states are separated by an energy barrier ΔE . In the context of magnetic systems and based on the stochastic Landau-Lifshitz-Gilbert equation, Brown has calculated an expression for the so-called dwell time $\tau = \tau_0 e^{\frac{\Delta E}{k_B T}}$ [W. F. Brown, Phys. Rev. 130, 1677 (1963)]. Here, both, the attempt time τ_0 as well as the energy barrier height ΔE are temperature dependent quantities. Generally, this formula is used to measure the barrier height ΔE from the temperature dependence of the dwelling time at low temperatures where ΔE is assumed temperature independent. Because of the macroscopic energy barriers and exponential temperature dependence, dwelling times are usually very large. In our case, however, as temperatures approach the critical temperature, the energy barrier vanishes, and τ_0 is then related to the frequency of the FM mode which - close to the critical temperature - is in the range of 50 GHz, an extremely high stochastic switching speed. To address this topic we added the following statement to the new chapter in the Supplementary Materials (“Mean dwell time τ estimation of the random telegraph noise”):

- We note that such thermally activated switching processes in a bistable systems where two states are separated by an energy barrier ΔE can be described by the Néel-Arrhenius law, which in the context of magnetic systems follows the equation $\tau = \tau_0 e^{\frac{\Delta E}{k_B T}}$ ⁵¹. Here, τ_0 is the attempt time, which is closely related to the intrinsic dynamics of the material system^{42,43}. Generally, this formula is used to measure the barrier height ΔE from the temperature dependence of the dwelling time at low temperatures where ΔE is assumed to be temperature independent. Because of the macroscopic energy barriers and exponential temperature dependence, dwelling times are usually very large. In our case, however, both ΔE and τ_0 are strongly temperature dependent and ΔE approaches zero at the critical temperature $T_{L,sim}$. Consequently, the mean dwell time τ_{sim} approaches the frequency of the F mode, which at this temperature is still in the range of 50 GHz (see Suppl. Fig. 4).
- As for the difference between the ferromagnets and antiferromagnets, the physics of the thermal switching itself is similar. Attempt frequencies, however, can be much higher in antiferromagnets because of their strongly different dispersion relations [V. Baltz, A.

Manchon, M. Tsoi, T. Moriyama, T. Ono, and Y. Tserkovnyak, Rev. Mod. Phys. 90, 015005 (2018); Levente Rózsa, Severin Selzer, Tobias Birk, Unai Atxitia, and Ulrich Nowak, Phys Rev. B 100, 064422 (2019)].

In our case, the switching process is not the 180 degree reversal of the antiferromagnetic Néel vector as discussed in the paper above but connected to the reorientation transition (see the Figure below). We would like to stress that this anisotropy-switching of the canted antiferromagnets in the *spontaneous* regime has not been discussed in the literature so far.

- While we believe that the physical picture of the presented switching between the anisotropy potential minima is explained thoroughly enough in the main text, we noticed that there was an expression which can potentially confuse the readers in the abstract. Therefore, we have made a revision as follows:

Abstract, main text:

- Before:
 “Owing to their high magnon frequencies, antiferromagnets are key materials for future high-speed spintronics. Picosecond switching of antiferromagnetic order has been viewed a milestone for decades and pursued only by using ultrafast external perturbations.”
- After:
 “Owing to their high magnon frequencies, antiferromagnets are key materials for future high-speed spintronics. Picosecond switching of antiferromagnetic **spin systems** has been viewed a milestone for decades and pursued only by using ultrafast external perturbations.”

Comment 3-5

In addition to above comments, I have following technical questions,

1. In SNS experiment, two color laser pulses are used and they have different wavelength may be 770 nm and 780 nm as written in the Methods. Can authors comment on why they use so close wavelength? Since femtosecond pulse laser with pulse width ~ 150 fs has bandwidth ~ 10 nm, I think that two laser pulses with close wavelength is difficult to be separated completely by the dichroic mirror.

- As long as the Faraday rotation experienced by the two probes are sufficiently large, it is in principle not impossible to drastically separate the two probe wavelengths. In our present case, choosing close wavelengths for the probe pulses had several technical advantages. First, the two focus spots had to perfectly overlap on the sample to obtain highest level of correlation. If the two wavelengths are too far apart, diffraction limited spot sizes could have a big difference and leads to the loss of the signal due to the poor spatial overlap. Second, since the Faraday coefficient, crystal birefringence and the absorption in the sample can differ for separate spectral regions, the balanced detection conditions could become significantly different in the two arms. For the most straightforward interpretation, as well as for the ease of alignment, we wanted to ensure that both probes experience the same amount of polarization rotation.
- As for the bandwidth, 150 fs is the pulse duration of the fundamental 1550 nm and not that of the second harmonics (SH) around 775 nm which we used for the probe. The bandwidth of the SH is around 3-4 nm, which is sufficiently narrow to allow full spectral separation by dichroic mirrors. Even though it negatively affects the temporal resolution, it is still sufficiently shorter compared with the relevant time scales of the magnon/switching dynamics reported here. We would like the Reviewer to refer to another comment (Comment 2-2), where we have added a revision in the **Methods** section in reply to a related question that has been asked by Reviewer 2.

Comment 3-6

2. In the calculation of resonance frequency is much higher than that obtained in the experiment [Fig. 4b and Fig. 4c]. What does this difference come from?

- This question about quantitative discrepancies between experiment and simulation has a similarity with comment 2.1 of referee #2, and we would like the reviewer to kindly refer to our reply to it. Generally said, the difference in the absolute value of the resonance frequency originates from the remaining imperfection of the optimization of the anisotropy parameters used in the calculation, which is technically unavoidable. We note that the numerical simulations of a reorientation transition based on an atomistic spin model is a very difficult task, with not too many successful examples published in the literature (one of which being our Ref.49, another one has just been published from: T. Dannegger, et al., Phys. Rev. B107, 184426 (2023)). The complications are manyfold. Leading energies in the simulated Hamiltonian are exchange interactions. They determine the Neel temperature,

while the DMI is responsible for the spin canting. The different anisotropy contributions determine the ground state spin orientation, the reorientation transition temperatures, the order of the phase transition, and its temperature range. But these anisotropy energies are two to three orders of magnitude smaller than the exchange energies and the reorientation transition is driven by a very delicate balance of the three different anisotropy energies, which are considered in our model. At the same time, all these parameters determine the resonance frequencies. We did our best to find a set of parameters that resembles our experimental findings as close as possible. We simulate already an enormous system size of 192^3 spins. To obtain a single time-dependent magnetization curve as shown in Figure 5a, a simulation run takes about 72 hours with a highly optimized code running on a graphics card. Slight modification of some of the model parameters might indeed lead to a better quantitative agreement with some of the experiments - but probably at the cost of others. It is practically impossible to find a set of only three anisotropy parameters, that fits simultaneously all experimental measurements. And even a "perfect" set of model parameters could not do that, since some deviations must be expected, since we cannot model all crystallographic details of our sample such as the effect of the inhomogeneity and the static magnetic domains in the experiment.

- To address this point and to the similar comment raised by Referee #2, we have revised the description of the simulation traces around Fig.4 in the main text and the corresponding part of **Methods** section. We would like to kindly ask Reviewer 3 to refer to our reply to Comment 2-1 for this revision.

Reviewers' Comments:

Reviewer #1:

Remarks to the Author:

the authors answered all my questions with care and diligence, and i recommend this article for publication.

Reviewer #2:

Remarks to the Author:

I am satisfied with the revisions, and recommend the publication of this paper as is.

Reviewer #3:

Remarks to the Author:

Authors revised the manuscript based on my previous comments. I am convinced by part of reply but I still do not understand some discussions about experimental spectra and author's explanations. I feel inconsistency throughout this manuscript. As I commented in a previous review, time-domain signal [Fig. 3a, 294.15 – 313.15 K] becomes broad when temperature is increased which corresponds to transition from ferromagnetic phase to antiferromagnetic phase. This corresponds to increase of correlation time or mean event time as authors added in the supplementary Fig. 8. This clearly indicates that ferromagnetic phase has shorter correlation time. In the abstract, authors stated that "The spectrum shows two distinct modes, one corresponding to the quasi-ferromagnetic mode and another one which has not been previously reported in pump-probe experiments. Comparison to a stochastic spin dynamics simulation reveals this new mode as smoking gun of ultrafast spontaneous spin switching within the double well anisotropy potential." I understood that this study is a first report of ultrafast spin noise spectroscopy of antiferromagnet or antiferromagnetic magnon mode in the other word. But I cannot understand that this new mode is related to ultrafast spontaneous spin switching. In addition to above comment, I understand from the reply that the signal peak width is coming from autocorrelation of magnon mode. Authors stated that the width of the autocorrelation mainly reflects the coherence time of the magnon dynamics. Then, it raises other questions why this coherence time of magnon is quite short? Why is the signal shape quite different with signals usually observed using pump-driven coherent magnon measurement? I read other ultrafast spin noise spectroscopy paper ref. 10. In ref. 10 they observed spin oscillation signal, but quite different with the results shown in this manuscript. One possible explanation is that the material authors studied may have large damping. This must be addressed.

I read other referee comments and thought that the novelty of this work may be measurement technique. Authors measured polarization rotation of two probe pulses first time to detect correlation of spin signals. For example, authors in ref. 10 detected polarization rotation of only one probe pulse. I think that it is better to mention what is the advantage of this method and what is difference with previous studies.

Reply Letter: Discovery of ultrafast spontaneous spin switching in an antiferromagnet by femtosecond noise correlation spectroscopy

Reviewer #1:

the authors answered all my questions with care and diligence, and i recommend this article for publication.

- We cordially thank the Reviewer for the ongoing support and for the positive assessment of our work.

Reviewer #2:

I am satisfied with the revisions, and recommend the publication of this paper as is.

- We cordially thank the Reviewer for the ongoing support and for the positive assessment of our work.

Reviewer #3:

Comment 1-1

Authors revised the manuscript based on my previous comments. I am convinced by part of reply but I still do not understand some discussions about experimental spectra and author's explanations. I feel inconsistency throughout this manuscript.

- We thank the Reviewer for the honest feedback. We hope to clarify our discussions and interpretation in the following.

Comment 1-2

As I commented in a previous review, time-domain signal [Fig. 3a, 294.15 – 313.15 K] becomes broad when temperature is increased which corresponds to transition from ferromagnetic phase to antiferromagnetic phase.

- We would like to kindly remind the reviewer that the spin reorientation transition (SRT) we study is not the phase transition between ferromagnetic and antiferromagnetic phases. Instead, the magnetic order is always in a canted antiferromagnetic phase within the investigated temperature regime. In the SRT, the orientation angle of the antiferromagnetic Néel vector, and consequently the net ferromagnetic moment, rotate by 90° within a finite temperature window, while the antiferromagnetic spin order itself remains unchanged. The SRT is widely observed in various magnetic systems that include transition-metal and rare-earth ions. The orthoferrite family is a typical, but not the only one of this class of materials.
- This context has been described in the original version of our manuscript as follows:

Our sample is a single crystal of the orthoferrite $\text{Sm}_{0.7}\text{Er}_{0.3}\text{FeO}_3$ ²⁴. In this material, the electron spins of the Fe^{3+} ions are antiferromagnetically coupled. A Dzyaloshinskii-Moriya interaction^{25,26} (DMI) results in a slight spin canting and a weak net ferromagnetic moment (net magnetisation \mathbf{M}). Orthoferrites have two exchange modes with resonance at multi-THz frequencies and two magnon modes in the sub-THz regime^{27,28}. The latter include a quasiferromagnetic mode (F mode) and a quasiantiferromagnetic mode (AF mode). The F mode is characterised by a precession of the weak ferromagnetic moment around its equilibrium axis, whereas the AF mode results in its longitudinal oscillation. $\text{Sm}_{0.7}\text{Er}_{0.3}\text{FeO}_3$ shows a temperature-induced second-order spin reorientation transition (SRT) close to room temperature²⁹. In the SRT region ($T_L < T < T_U$), \mathbf{M} continuously rotates from along the crystallographic a -axis at $T_L = 310$ K to the c -axis at $T_U = 330$ K²⁴.

- To further clarify, we have revised the text as follows:

Sample description, around Fig 1b:

- o Before:
“reorientation transition (SRT) close to room temperature²⁹, in which the magnetic anisotropy (but not the antiferromagnetic spin order) change. In the SRT region ($T_L < T < T_U$), \mathbf{M} continuously rotates from along the crystallographic a -axis at the

lower critical temperature T_L to the c-axis at the upper critical temperature T_U . The critical temperatures have been previously measured to be $T_{L, \text{ref}} = 310 \text{ K}$ and $T_{U, \text{ref}} = 330 \text{ K}$ ²⁴.”

- After:
“ $\text{Sm}_{0.7}\text{Er}_{0.3}\text{FeO}_3$ shows a temperature-induced second-order spin reorientation transition (SRT) close to room temperature²⁹, in which the magnetic anisotropy (but not the antiferromagnetic spin order) change. In the SRT region ($T_L < T < T_U$), \mathbf{M} continuously rotates from along the crystallographic a -axis at the lower critical temperature T_L to the c-axis at the upper critical temperature T_U . The critical temperatures have been previously measured to be $T_{L, \text{ref}} = 310 \text{ K}$ and $T_{U, \text{ref}} = 330 \text{ K}$ ²⁴.”

Comment 1-2

This corresponds to increase of correlation time or mean event time as authors added in the supplementary Fig. 8. This clearly indicates that ferromagnetic phase has shorter correlation time.

In the abstract, authors stated that “The spectrum shows two distinct modes, one corresponding to the quasi-ferromagnetic mode and another one which has not been previously reported in pump-probe experiments. Comparison to a stochastic spin dynamics simulation reveals this new mode as smoking gun of ultrafast spontaneous spin switching within the double well anisotropy potential.” I understood that this study is a first report of ultrafast spin noise spectroscopy of antiferromagnet or antiferromagnetic magnon mode in the other word. But I cannot understand that this new mode is related to ultrafast spontaneous spin switching.

- As discussed in the main text, there are indeed two distinct features in both the observed waveforms and their corresponding spectra. One is the relatively high frequency and strongly temperature-dependent oscillation with a period of 10 to 20 ps, which is unambiguously assigned to the quasiferromagnetic mode (F mode) because the temperature dependence of its frequency precisely matches the reported quantity of pump-probe results. The other one, which we call low-frequency (LF) mode, is the new mode which is now under question by the Referee.
- We assume that the Referee’s concern is how this LF mode should be interpreted as a spin switching. As described thoroughly in the main text, we assigned this by comparing the spectral features of the experiment with that of the theoretical simulation. In the simulated spectra the LF mode also appears aside from the F mode, and in the corresponding time-domain waveforms (not the autocorrelation but the real-time evolution of the magnetisation (Fig.5)), the bistable behaviour characteristic to the random switching is observed.
- As explained in the Supplementary Material, the autocorrelation of the bistable switching dynamics corresponds to an exponential decay feature around $t = 0$. This exponential decay feature is clearly observed in the original experimental autocorrelation waveforms, e.g., 313.15 K

in Fig. 3a. This fact represents direct evidence that the LF mode originates from the switching dynamics. We additionally note that at 313.15 K, a faster oscillation is seen around the peak. This corresponds to the F mode within a single potential well. Coexistence of the switching and the F mode is thus directly visible in the autocorrelation waveform at this temperature. At lower temperatures, e.g. at 303.15 K, the exponential component is not as clear as at 313.15 K because the decay time is much shorter and the amplitude is smaller. Still, in the frequency domain they are clearly distinguishable as two separate spectral peaks (Fig. 4a), as described in detail in the main text.

- In the lower-temperature side of the SRT region, the exponential decay time (mean event time) τ becomes shorter. This is because the anisotropy barrier in the potential separating the two minima becomes smaller around the SRT critical threshold temperature T_L (around 304~305 K). At the same time, a small potential barrier directly means that the angular separation of the two minima position is also small. Therefore, its contribution to the autocorrelation amplitude is much smaller in this region compared to the higher temperatures (e.g., 313.15 K), where the angular separation is much larger. We interpret that this is why the strongest switching signal is observed at temperature slightly higher than T_L . Still, the switching phenomena persists down to T_L , where the mean event time decreases to $\tau = 11$ ps, as discussed in the Supplementary Material.
- This comment gave us the awareness that some of the expressions originally used in the main text may be potentially confusing. Therefore, we have revised the text as follows:

Description of Figure 3a in the main text:

- o Before:

“ Figure 3a depicts the striking variation of spin noise autocorrelation waveforms found around the SRT. The amplitudes, oscillation periods and lifetimes depend strongly on temperature. Since in the SRT only the spin system changes, this strong variation of the signal indicates that the observed autocorrelation signal truly comes from magnon noises. The width of the autocorrelation around the peak ($\Delta t = 0$) mainly reflects the coherence time of the magnon dynamics. The oscillations with period of tens of picosecond clearly visible in some of the autocorrelation waveforms (e.g., 294.15 K and 303.15 K) on both sides of the peak reflects the spin precession mode. The temporal dynamics is discussed later in more detail.

Now, we first focus on the temperature evolution of the signal amplitude. The amplitude at $\Delta t = 0$ is depicted in Fig. 3b. A sharp amplitude increase is observed in the region around 312.15 K. This point is slightly higher but close to the estimated lower threshold temperature $T_L \sim 305$ K of the SRT in our sample, around which temperature the anisotropy softening results in an enhanced magnetic susceptibility²⁷. This close coincidence indicates that the amplitude of the observed magnon noise can be naively understood as the angle distribution of spins due to thermal occupation of the anisotropy potential well by magnons, consistent with the model described in Fig. 1b. Beyond this temperature, the noise amplitude decreases continuously, almost disappearing around the upper threshold $T_U = 320$ K. The sharp decrease observed towards the higher temperature side is explained by the equilibrium rotation of the spin system within the SRT. In this temperature region, the net magnetisation M continuously

rotates from $\mathbf{M} // a$ (in-plane) to $\mathbf{M} // c$ (out-of-plane). Our Faraday probe is sensitive only to the c -axis magnetisation fluctuation δM_c of the F mode [**Supplementary Materials**], which is expected to reduce as $\delta M_c \propto \cos(\theta)$ towards higher temperature. Therefore, the noise amplitude is expected to decrease continuously. ”

○ After:

“ Figure 3a depicts the striking variation of spin noise autocorrelation waveforms found around the SRT. The amplitudes, oscillation periods and lifetimes depend strongly on temperature. Since in the SRT only the **orientation of the** spin system changes, this strong variation of the signal indicates that the observed autocorrelation signal truly comes from **magnetisation noise dynamics. The finite width of the autocorrelation around the peak ($\Delta t = 0$) indicates that the system exhibits non-zero coherence on the ultrafast timescale.** The oscillations with period of tens of picosecond clearly visible in some of the autocorrelation waveforms (e.g., 294.15 K and 303.15 K) on both sides of the peak reflects the spin precession mode. The temporal dynamics is discussed later in more detail.

Now, we first focus on the temperature evolution of the signal amplitude. The amplitude at $\Delta t = 0$ is depicted in Fig. 3b. A sharp amplitude increase is observed in the region around 312.15 K. This point is slightly higher but close to the estimated lower threshold temperature $T_L \sim 305$ K of the SRT in our sample, around which temperature the anisotropy softening results in an enhanced magnetic susceptibility²⁷. Beyond this temperature, the noise amplitude decreases continuously, almost disappearing around the upper threshold $T_U = 320$ K. The sharp decrease observed towards the higher temperature side is explained by the equilibrium rotation of the spin system within the SRT. In this temperature region, the net magnetisation \mathbf{M} continuously rotates from $\mathbf{M} // a$ (in-plane) to $\mathbf{M} // c$ (out-of-plane). Our Faraday probe is sensitive only to the c -axis magnetisation fluctuation δM_c of the F mode [**Supplementary Materials**], which is expected to reduce as $\delta M_c \propto \cos(\theta)$ towards higher temperature. Therefore, the noise amplitude is expected to decrease continuously. **These temperature dependencies indicate that the amplitude of the observed magnon noise can be naively understood as the angle distribution of spins due to thermal occupation of the anisotropy potential well by magnons, consistent with the model described in Fig. 1b.** ”

To avoid misinterpretation of the low-frequency feature as a magnon mode, we also replaced “LF mode” with “**LF feature**” or “**LF peak**” throughout the manuscript.

Comment 1-3

In addition to above comment, I understand from the reply that the signal peak width is coming from autocorrelation of magnon mode. Authors stated that the width of the autocorrelation mainly reflects the coherence time of the magnon dynamics. Then, it raises other questions why this coherence time of magnon is quite short? Why is the signal shape quite different with signals usually observed using pump-driven coherent magnon measurement?

- As already described in the reply to Comment 1-2, the autocorrelation trace is made up of two distinct features. On one hand, the incoherent noise of the F mode and on the other hand, the signature of stochastic switching. At sufficiently low temperatures (e.g., 294.15 K), where no switching occurs, the width of the autocorrelation reflects solely the coherence time of the F mode magnon. In the SRT regime, however, the width of the autocorrelation is determined by the convolution of the two mechanisms and is eventually dominated by the exponentially decaying signature of the ultrafast switching towards the higher temperatures (peaking around 313 K).
- As for the coherence time of the magnon dynamics: in this material, the F mode lifetime is known to be in the order of 20 – 40 ps around the SRT region. This lifetime is previously confirmed quantitatively by a regular pump-probe experiment [G. Fitzky et al. PHYSICAL REVIEW LETTERS **127**, 107401 (2021)]. This value is reasonable compared to the lifetimes generally observed for the F mode near the SRT region in orthoferrites [e.g., J. A. et al., Phys. Rev. B **84**(10), 104421 (2011)]. Therefore, the coherence time of the F mode measured in our autocorrelation is consistent with the original magnon dynamics.

Comment 1-4

I read other ultrafast spin noise spectroscopy paper ref. 10. In ref. 10 they observed spin oscillation signal, but quite different with the results shown in this manuscript. One possible explanation is that the material authors studied may have large damping. This must be addressed.

- Note that in Ref [10], a paramagnetic spin system was investigated, namely Larmor precession of free electrons in GaAs under high external magnetic field. Paramagnets usually tend to exhibit quite long lifetime because they do not constitute magnons and therefore the magnon-magnon scattering and magnon-phonon interactions, which would be present in the case of correlated magnets and lead to the shortening of the magnon lifetime, are absent.
- It should be noted that our sample $\text{Sm}_{0.7}\text{Er}_{0.3}\text{FeO}_3$ has a relatively short magnon lifetime in the first place, as can be seen even in the pump-probe data (see our reply Comment 1-3).

Comment 1-5

I read other referee comments and thought that the novelty of this work may be measurement technique. Authors measured polarization rotation of two probe pulses first time to detect correlation of spin signals. For example, authors in ref. 10 detected polarization rotation of only one probe pulse. I think that it is better to mention what is the advantage of this method and what is difference with previous studies.

- We thank the Reviewer for this suggestion. We would like to note that, even in Ref 10, two probe pulses are used for sampling the autocorrelation. However, the two probes were detected with one single balanced photodetector. This is the key difference from our case since we now detect the two probe pulses with two separate BPDs.
- This is in fact a very important difference. When a single photodetector is hit twice by fs pulses, the first pulse introduces photocarriers that accumulate for a finite time, typically ~ 100 ps for a fast Si PIN photodiode, thus reducing the absorption for the second pulse due to saturated absorption. This nonlinearity in the photodiode directly leads to spurious signals and therefore a loss of sensitivity in a time window of approximately 100 ps around $t = 0$ in the autocorrelation trace. Therefore, in the scheme where the two probe pulses are measured by a single photodetector, the minimum delay time of the autocorrelation is inherently linked to the optical nonlinearity of the detector itself.
- This is exactly why we used two separate balanced photodetectors for the two probe pulses. This goal is enabled by tuning them to slightly different wavelengths and splitting the optical signal with a dichroic filter after the sample. By this way, each photodiode is hit only once by a fs probe

within an event, thereby eliminating the influence of the optical nonlinearity of the diode in the data and therefore enabling to fully cover the time window all the way down to $t = 0$ ps in the autocorrelation trace. As discussed above, this ability is crucial to investigate correlated spin systems with a short magnon coherence lifetime.

- This point is already mentioned in the main text, but we noticed that it was only explained in passing. Therefore, we have revised and added the following sentences to the end of the “Experiment and data post-processing” part of the “Methods” section:

“Contrary to previous work^{10,12}, the two-colour scheme also allows us to sample the Faraday rotation with two separate BPDs. By this way, our time resolution is not limited by the deadtime of the photodetector and we can exploit the full sub-picosecond resolution given by the duration of the probe pulses. This fact enables us to analyse the changes of spin noise over a magnetic phase transition in a correlated magnetic material with significantly shorter spin dephasing times.”

In addition to the above revisions, we have also applied minor revisions throughout the main text. Furthermore, we added the following revision which addresses a minor difference to the model described in the main text:

End of Supplementary Material:

Before:

“Furthermore, we find dwell times as low as $\tau = 11.21$ ps for the lowest evaluated temperature of 304.35 K indicating that indeed our investigated system shows picosecond random switching.”

After:

“Furthermore, we find dwell times as low as $\tau = 11.21$ ps for the lowest evaluated temperature of 304.35 K indicating that indeed our investigated system shows picosecond random switching. It is interesting to note that the signature of the RTN persists down to temperatures slightly lower than the estimated critical threshold temperature $T_L \sim 305$ K. We assume that this is due to the slight variation of T_L by the local magnetic environment within the probe spot due to, e.g., static magnetic domains.”

Reviewers' Comments:

Reviewer #3:

Remarks to the Author:

Authors fully addressed the points I mentioned in the previous review. I am convinced by the author's explanations.

Now I recommend that the paper is published as is.